# Cohesin residency determines chromatin loop patterns

**Lorenzo Costantino†, Tsung-Han S Hsieh†, Rebecca Lamothe†, Xavier Darzacq, Douglas Koshland\***

Department of Molecular and Cell Biology, University of California, Berkeley, Berkeley, United States

**Abstract** The organization of chromatin into higher order structures is essential for chromosome segregation, the repair of DNA-damage, and the regulation of gene expression. Using Micro-C XL to detect chromosomal interactions, we observed the pervasive presence of cohesin-dependent loops with defined positions throughout the genome of budding yeast, as seen in mammalian cells. In early S phase, cohesin stably binds to cohesin associated regions (CARs) genome-wide. Subsequently, positioned loops accumulate with CARs at the bases of the loops. Cohesin regulators Wpl1 and Pds5 alter the levels and distribution of cohesin at CARs, changing the pattern of positioned loops. From these observations, we propose that cohesin with loop extrusion activity is stopped by preexisting CAR-bound cohesins, generating positioned loops. The patterns of loops observed in a population of wild-type and mutant cells can be explained by this mechanism, coupled with a heterogeneous residency of cohesin at CARs in individual cells.

**\*For correspondence:**
koshland@berkeley.edu

†These authors contributed equally to this work

**Competing interests:** The authors declare that no competing interests exist.

## Introduction

The genome is organized into three-dimensional structures that are essential for the correct segregation of chromosomes, the efficient repair of DNA damage, and the regulation of gene transcription. The initial existence of higher order chromosome structures was discovered in the early 1900s by the dramatic cytological appearance of pairs of condensed sister chromatids in mitosis. Moreover, electron microscopy studies showed that mitotic and meiotic chromatin is organized in loops with a protein scaffold (*Gall and Callan, 1962*; *Marsden and Laemmli, 1979*; *Paulson and Laemmli, 1977*).

A high-resolution view of chromosome folding became possible with the development of new methods based on chromosome conformation capture (3C) technology (*Dekker et al., 2002*; *Denker and de Laat, 2016*; *Kempfer and Pombo, 2020*; *Lieberman-Aiden et al., 2009*). Essentially, 3Cs methods stabilize the interactions between different regions of the chromatin that are in close spatial proximity, detect these interactions, and visualize them on a contact map. These methods showed in mammalian cells the existence of chromatin loops at specific loci (*Deng et al., 2012*; *Palstra et al., 2003*; *Tolhuis et al., 2002*). Later papers revealed defined loops at defined positions genome-wide that persisted from interphase to the onset of preprophase but not later in mitosis (*Rao et al., 2014*). Moreover, 3Cs methods showed that mammalian chromosomes are organized into self-associating domains, which contain sequences that are more likely to interact with each other than with the rest of the genome (*Dixon et al., 2012*; *Nora et al., 2012*; *Sexton et al., 2012*). The existence of these loops has inspired several outstanding questions, including what are the molecular mechanisms that drive the formation of loops, their genome distribution, their frequency in a population of cells, and their biological function(s).

A major advance came with the discovery that loops formation depended upon the evolutionarily conserved SMC (structural maintenance of chromosomes) complexes (*Gassler et al., 2017*; *Rao et al., 2017*; *Sanborn et al., 2015*; *Schwarzer et al., 2017*; *Wang et al., 2017*; *Wutz et al.,*

2017). SMC complexes can tether together two regions of DNA, either within a single DNA molecule or between DNA molecules (*Hassler et al., 2018*; *Onn et al., 2008*). Additionally, they can translocate along DNA to extrude DNA loops (loop extrusion) in vitro and in vivo (*Davidson et al., 2019*; *Ganji et al., 2018*; *Kim et al., 2019*). Loop formation by one SMC complex, called cohesin, has been extensively studied in mammalian cells (*Hadjur et al., 2009*; *Parelho et al., 2008*; *Wendt et al., 2008*). The CTCF DNA-binding factor defines the position of most of the cohesin-dependent loops (*Handoko et al., 2011*; *Li et al., 2012*; *Splinter et al., 2006*). In fact, cohesin-dependent loops occur preferentially between the two CTCF-binding sites that are in a convergent orientation (*Guo et al., 2015*; *Rao et al., 2014*; *de Wit et al., 2015*). Cohesin-dependent loops occurred between more distal CTCF-binding sites when the cohesin regulator WAPL was depleted, showing that WAPL restricts loop size. WAPL dissociates cohesin from chromosomes (*Haarhuis et al., 2017*), and by preventing WAPL binding to cohesin, CTCF is thought to stabilize cohesin and prevent loop disruption (*Li et al., 2020*). The formation of specific loops by cohesin, CTCF, and WAPL has been proposed to contribute to the regulation of gene expression by facilitating the interactions of enhancers with distal promoters (*Schoenfelder and Fraser, 2019*).

However, five observations paint a more complicated picture of the regulation and function of cohesin-dependent looping. First, robust loops still form when the orientation preference of CTCF sites was eliminated through depletion of a cohesin regulator (*Wutz et al., 2017*). Clearly, CTCF orientation is not the only determinant of loop position in mammals. Indeed, looping is observed in organisms that do not have the CTCF factor (*Heger et al., 2012*; *Rowley et al., 2020*). Thus, the fundamental mechanism(s) of positioning loops in most organisms including mammals remains to be determined. Second, a cohesin auxiliary factor, PDS5, facilitates WAPL's ability to remove cohesin from chromosomes, yet the impact of WAPL and PDS5 on looping appear to be different (*Wutz et al., 2017*). This observation suggests that cohesin regulators control looping by additional yet to be discovered mechanisms. Third, the current understanding of cohesin, CTCF, WAPL, and PDS5 does not fully explain why the intensity of each loop can vary dramatically in a cell population, and how the loop pattern changes upon the inactivation of cohesin regulators (*Haarhuis et al., 2017*; *Wutz et al., 2017*). Fourth, while there are studies that analyzed cohesin residency and the turnover on DNA (*Fudenberg et al., 2016*; *Wutz et al., 2020*), it is still unclear how they exactly contribute to loop formation. Fifth, the postulated function of looping in gene expression does not explain why most regions of the mammalian genome have cohesin-dependent loops, yet the proper expression of most genes is cohesin/CTCF independent (*Nora et al., 2017*; *Rao et al., 2017*; *Schwarzer et al., 2017*). These observations show that key aspects of the molecular mechanisms and the biological functions of looping remain to be elucidated.

A deep understanding of complex cellular processes often resulted from their study in diverse organisms. Indeed loop formation in yeast was analyzed with a high-throughput 3C-method called Hi-C. Interestingly, a seminal study showed that cohesin mediates a change in chromosomal interactions during mitosis that, through polymer simulations, is compatible with the presence of chromatin loops (*Schalbetter et al., 2017*). Another study showed that loop size was restricted by Wpl1p (yeast ortholog of WAPL) as observed in mammalian cells, and by Pds5 (yeast ortholog of PDS5) (*Dauban et al., 2020*). However, the signal for loops with a defined position appeared sporadically in the genome and was poorly defined when compared to mammalian interphase cells (*Dauban et al., 2020*; *Paldi et al., 2020*; *Schalbetter et al., 2017*). The weak signal for positioned loops in yeast might have been a reflection of a biological difference between yeast and mammalian cells, hinted by the absence of a CTCF protein in yeast, or methodological limitations of these studies. A recent study revealed a robust signal for cohesin-dependent positioned loops during meiosis (*Schalbetter et al., 2019*), revealing that yeast has indeed the capability to efficiently form positioned loops.

We reasoned that the formation of positioned loops in yeast could be cell cycle-regulated. Moreover, given the small size of the yeast genome relative to mammals, positioned loops in yeast could be smaller and their detection might require a high-resolution methodology. Micro-C and its evolution Micro-C XL can detect chromatin interactions from the scale of nucleosomes to the whole genome (*Hsieh et al., 2015*; *Hsieh et al., 2016*). Indeed, this method successfully detected positioned loops in quiescent cells and in S phase of mitotically dividing yeast (*Ohno et al., 2019*; *Swygert et al., 2019*). Here, we use Micro-C XL on synchronized populations of yeast cells to improve the detection of chromatin loops and observe a robust signal for positioned loops genome-

wide. The analysis of this loop signal led us to a model that explains the genomic distribution and frequency of loops and their relationship to cohesin binding to DNA.

## Results

### Micro-C XL reveals prevalent chromatin loops with a defined position in mitotic chromosomes

To maximize the detection of loops, we arrested cells in mitosis, using the inhibitor of microtubule polymerization nocodazole (Nz), where chromosomes are highly structured and analyzed them by high-resolution Micro-C XL analysis with improved short-range interactions detection (*Figure 1A*, *Supplementary file 1*, and *Hsieh et al., 2015*; *Hsieh et al., 2016*). In brief, we treated cells with crosslinkers to preserve existing contacts between nucleosomes and then digested the crosslinked chromatin into mono-nucleosomes using the micrococcal nuclease (MNase). The free nucleosomal DNA ends were labeled, ligated, isolated, and subjected to paired-end sequencing. The paired reads from the chimeric sequences gave the coordinates of nucleosomes that were contacting in vivo. We plotted these coordinates producing a contact map of all the interacting nucleosomes in a cell population (*Figure 1A*), generating a matrix of ~66,000 × 66,000 nucleosomes/pixels for the yeast genome.

Three primary 3D features of chromosomes can be found on contact maps. First, local interactions between neighboring nucleosomes are likely, giving rise to a robust signal along the diagonal on the contact map (*Figure 1B*, unstructured). Second, in a chromatin domain, nucleosomes within the domain will interact with each other more frequently than ones in the neighboring domains, giving rise to a square signal on the contact map (*Figure 1B*, interaction domain). Third, a positioned loop is made when the same pair of distal regions (5' and 3' loop anchors) are bridged together with enough frequency in a cell population. The interacting nucleosomes from the loop anchors will generate an off-diagonal spot/focal interaction on the contact map (*Figure 1B*, loop). The positioned loop size is equal to the length of DNA between the loop anchors and is equal to the distance of the spot from the diagonal. Loops that are randomly located in the genome of a cell population are not immediately visualized in the contact map. Previous papers have inferred the presence, size, and genome distribution of randomly located loops through polymer modeling and simulation (*Schalbetter et al., 2017*).

Micro-C XL analysis of mitotically arrested cells revealed many off-diagonal spots on the contact map, indicative of the presence of positioned loops. These positioned loops became easily visible when zooming-in on smaller regions of the chromosome with increasing resolution (*Figure 1C*, top row). The pervasive presence of positioned loops shown on chromosome 7 was observed in the entire genome. These positioned loops were less frequent and less defined in contact maps using datasets from previously published Hi-C maps of mitotic cells (*Figure 1C*, bottom rows; *Figure 1— figure supplement 1A*) and Micro-C XL analysis of asynchronous cells (*Figure 1—figure supplement 1B*). The improved detection of positioned loops in Micro-C XL was likely due to its superior resolution in the range from 100 bp to 30 kb (*Figure 1D*), as shown in the genome-wide analysis of interaction decaying rate. We used two different loop-calling programs, HiCCUPS and Chromosight, to identify the location of positioned loops genome-wide (*Durand et al., 2016*; *Matthey-Doret et al., 2020*). Both identified 10–20 times more positioned loops in Micro-C XL than Hi-C (*Supplementary file 2*). The total number of 3' and 5' loop anchors for positioned loops was also 10 times greater in Micro-C XL than Hi-C (*Figure 1E*). We piled-up the signal from the contact map of a 5 kb region with each positioned loop spot at its center. We compared the center (positioned loop signal) versus the four edges (background) as a ratio and detected a ~4 fold increase in Micro-C XL (4 vs 1), but only a minor increase in Hi-C (1.2 vs 1) (*Figure 1F*). This strong signal for positioned loops allowed us to locate precisely the chromosomal position of their loop anchors genome-wide. Thus, Micro-C XL is a robust assay to detect chromosome loops, suggesting that positioned loops are a ubiquitous folding feature of yeast mitotic chromosomes.

### Cohesin complex mediates loop formation

We proceeded to identify the protein/s that are responsible to form the positioned loops in yeast mitotic chromosomes. In mammals, cohesin is required for loop formation and is present at loop

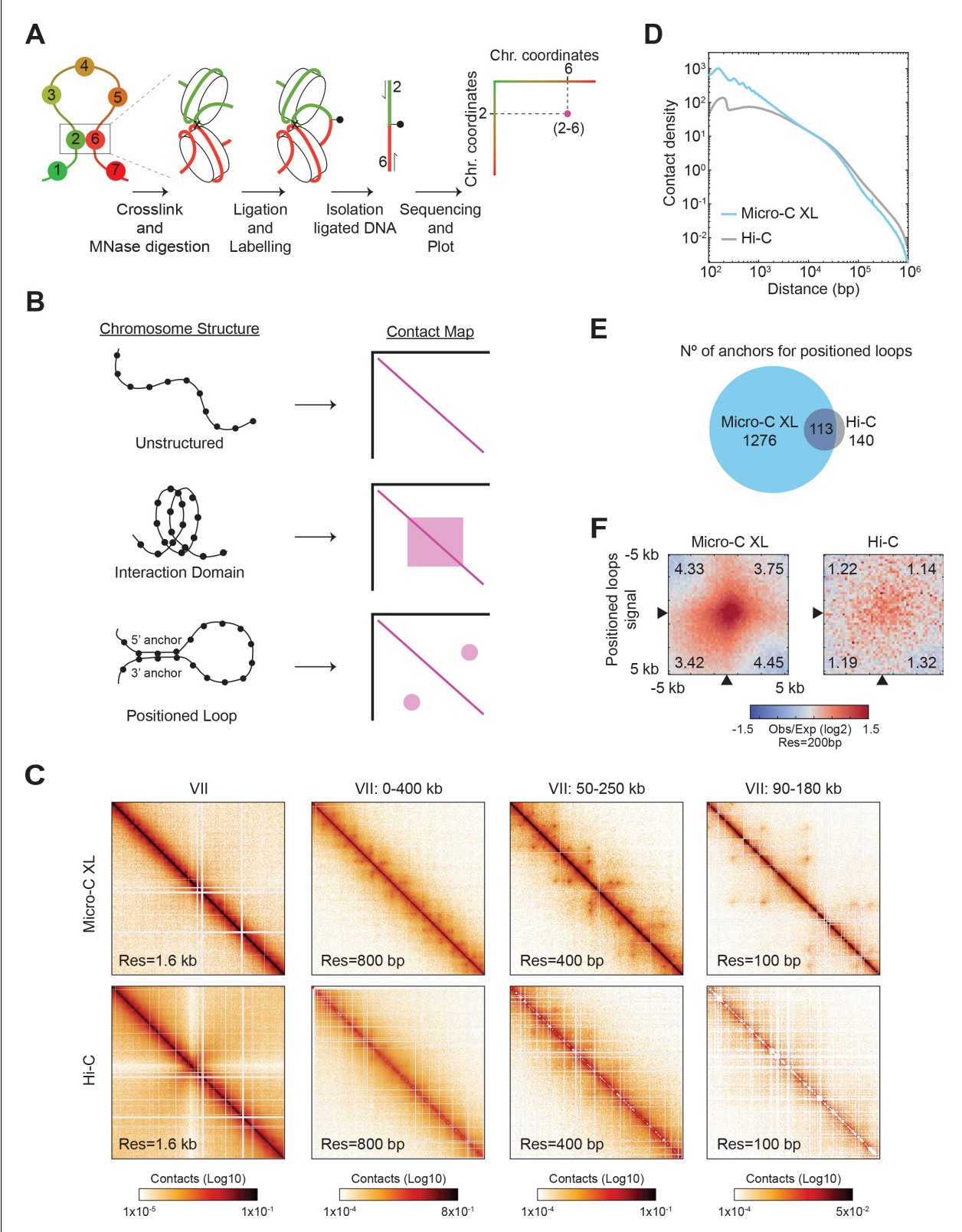

**Figure 1.** Micro-C XL reveals positioned loops in mitotic chromosomes. (A) Micro-C XL maps chromatin conformation by detecting nucleosomes that interact in vivo. Schematic illustrates how two interacting nucleosomes (e.g. 2 and 6) are visualized on the contact map. (B) Illustration of different chromosomal structures and their corresponding signal on the contact map. Unstructured chromosomes: only neighboring nucleosomes will be crosslinked, producing a linear signal along the diagonal. Interaction Domain: the nucleosomes within a domain will be also crosslinked, forming a

*Figure 1 continued on next page*

*Figure 1 continued*

square along the diagonal. Positioned loop: the nucleosomes at the base of the loop (5' and 3' anchors) will be crosslinked, forming a spot away from the diagonal. These different structures can form concomitantly on chromosomes producing contact maps with squares and spots along the diagonal. (C) Comparison of contact maps from mitotically arrested cells produced with Micro-C XL (top) and Hi-C (bottom) (*Schalbetter et al., 2019*). Contact maps show successive zoom-ins of chromosome VII (from the full 1090 kb to a 90 kb region), across multiple resolutions (from 1.6 kb to 100 bp). All contact maps throughout the manuscript have the standard colormap scheme that uses the shades of red from white (no detectable interactions) to black (maximum interactions detected) in log10 scale. (D) Micro-C XL uncovers chromatin interactions below the kilobase range. Interactions-versus-distance decaying curve shows the normalized contact density (y-axis) against the distance between the pair of crosslinked nucleosomes from 100 bp to 1 Mb (x-axis) for Micro-C XL and Hi-C (*Schalbetter et al., 2019*). (E) Micro-C XL detects abundant mitotic positioned loops and their corresponding loop anchors. Venn diagrams show how Micro-C identifies 1276 anchors for defined loops that include most of the loop anchors identified by Hi-C. (F) Genome-wide average heatmaps show a sharp enrichment of positioned loop signal in Micro-C XL data. Heatmaps were plotted using the 200 bp resolution data for Micro-C XL and Hi-C. The plot is a piled-up of the ±5 kb region surrounding the loop anchors (black arrows) of every loop identified in Micro-C XL. Numbers in the corners represent the fold-change of the signal enrichment of the center pixel over the indicated corner pixels ($14^2$ pixels). The contact map was normalized by matrix balancing and distance, using a diverging colormap with positive signal enrichment in red and negative signals in blue (gradient of normalized contact enrichment is in log2).

The online version of this article includes the following figure supplement(s) for figure 1:

**Figure supplement 1.** Comparison of Micro-C XL and Hi-C mapping in budding yeast.

anchors (*Gassler et al., 2017*; *Rao et al., 2017*; *Sanborn et al., 2015*; *Schwarzer et al., 2017*; *Wang et al., 2017*; *Wutz et al., 2017*). In budding yeast, cohesin is needed for loops genome-wide (*Dauban et al., 2020*; *Schalbetter et al., 2017*). To assess if cohesin was also needed for the formation of positioned loops in budding yeast, we first identified the genomic positions of cohesin binding by ChIP-seq of the Mcd1p cohesin subunit (also known as Scc1p/Rad21p) in mitotically arrested cells. Cohesin binds to discrete chromosomal loci termed cohesin associated regions (CARs), which are distributed along chromosomal arms preferentially between terminators of two convergently transcribed genes, and at centromeres (*Blat and Kleckner, 1999*; *Glynn et al., 2004*; *Laloraya et al., 2000*; *Megee and Koshland, 1999*). By aligning the Mcd1p ChIP-seq signal with the Micro-C XL contact map, we observed that the position of every positioned loop anchor overlapped with the position of a CAR (*Figure 2A*). To evaluate the robustness of this correlation genome-wide, we plotted the Mcd1p ChIP-seq signal of every sequence involved in a positioned loop (the 5' and 3' anchors, and the region in between), by aligning their 5' and 3' anchors (*Figure 2B*). We observed an enrichment of Mcd1p over >98% of positioned loop anchors. Therefore, cohesin associated regions and the anchors for positioned loops are highly correlated in mitotic chromosomes of budding yeast, suggesting that CARs help define the position of loops.

To gather more information on the portion of the CAR that functions as a loop anchor, we compared the signals from the contact map with the cohesin ChIP-seq. For this analysis, we piled-up the contact map signal of the five kilobases regions centered at the intersection of the coordinates of each pair of Mcd1p ChIP-seq peaks (*Figure 2C*). The portion of the contact map with a contact probability greater than 90%, corresponded to a 1 kilobase region that was centered at the summit of the Mcd1p ChIP-seq peak. Coincidentally, the average width of Mcd1p ChIP-seq peaks is also ~1 kilobase. Thus, anchors for positioned loops in budding yeast are centered at the peak of cohesin binding at CARs and both are confined within 1 kilobase region. These results further correlate anchors for positioned loops with cohesin associated regions on mitotic chromosomes.

This correlation prompted us to examine the causal role of cohesin in positioned loop formation. To test this possibility, we depleted the cohesin subunit Mcd1p using an auxin-induced degron system (*Figure 2—figure supplement 1A* and *Eng et al., 2014*) and analyzed loop formation by Micro-C XL. Without cohesin, we detected no distinguishable positioned loops by visual inspection of the contact map (*Figure 2D* and *Figure 2—figure supplement 1B*). The positioned loops were also gone genome-wide using the loop calling program, dropping from 732 in wild-type cells to just five when cohesin was depleted (*Figure 2E*). Finally, we analyzed the genome-wide average signal coming from the piled-up of 5 kilobases regions centered on each positioned loop spot (*Figure 2F*). The signal for the positioned loop spots was reduced to background upon Mcd1p depletion. We conclude that cohesin is required for positioned loop formation genome-wide.

Condensin, another member of the SMC family, was known to promote mitotic chromosome structure by condensing chromosomes in vertebrate and yeast cells (*Hirano, 2016*; *Hirano and*

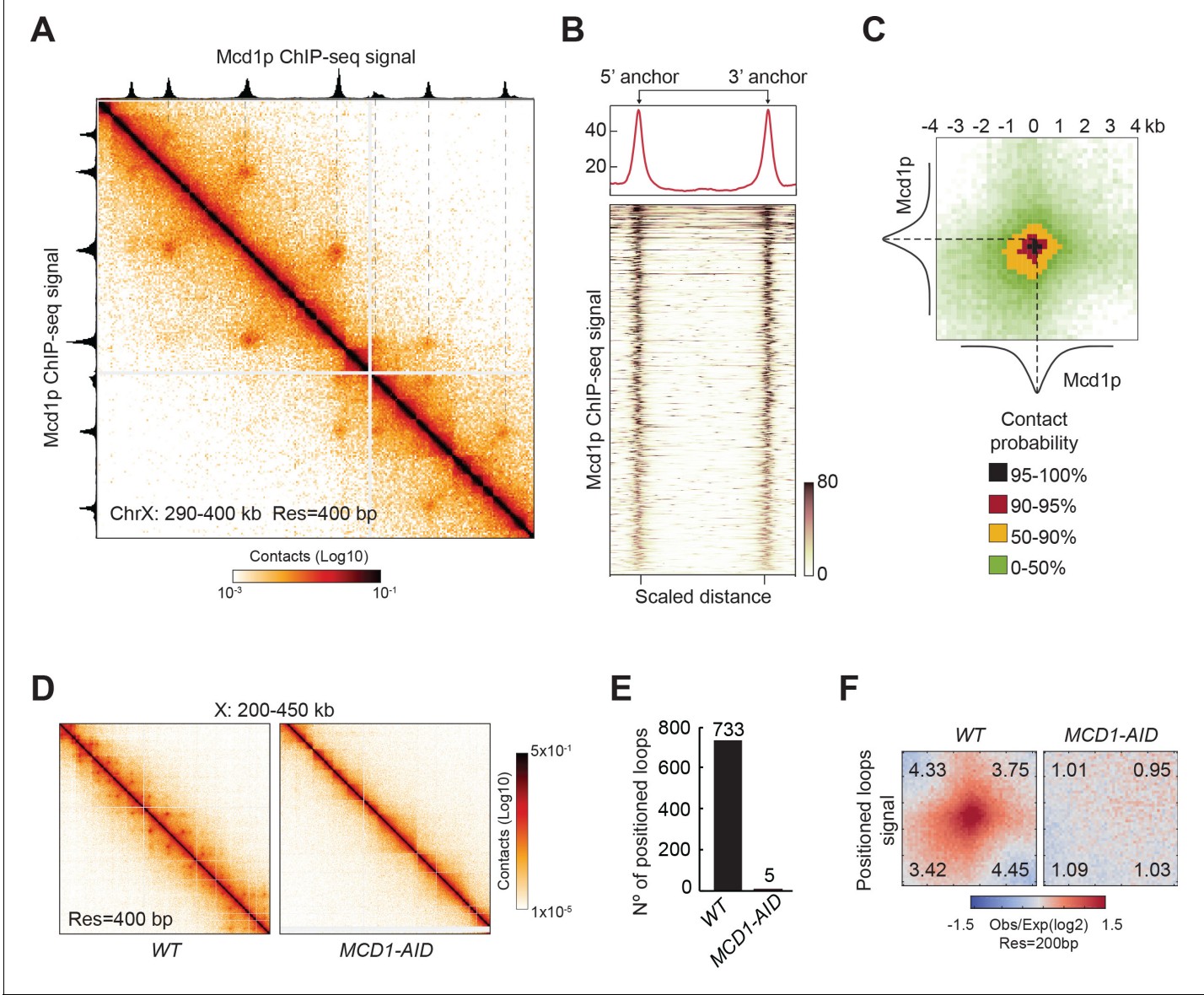

**Figure 2.** Mitotic positioned loops are cohesin-dependent. (**A**) Cohesin peaks colocalize with the loop anchors of positioned loops. Contact map showing the interactions in the 290–400 kb region of chromosome X overlays with Mcd1p ChIP-seq tracks on top and left. The dashed lines indicate the colocalization of anchors of positioned loops and Mcd1p peaks. (**B**) Cohesin binding is enriched at anchors for positioned loops genome-wide. Heatmap (bottom panel) shows the enrichment of the Mcd1p ChIP-seq reads at the loop anchors genome-wide (732 rows). Pairs of loop anchors were rescaled to the same length. Genome-wide average ChIP-seq signal over the rescaled loop anchors is plotted (top panel). (**C**) Mcd1p peaks center on the anchors for positioned loops. Heatmaps were plotted with 200 bp resolution of contact map signal in ±4 kb regions surrounding the paired Mcd1p peaks. The contact map is colored by the contact probability with >95% in black, 90–95% in red, 50–90% in yellow, 0–50% in shades of green. The average Mcd1p peak is overlaid with contact maps and centered at the loop bases. (**D**) The positioned loop signal on the contact map is lost upon cohesin depletion. Contact maps for *WT* and *MCD1-AID* Micro-C data show substantial loss of loop signal over chromosome X: 200–450 kb upon Mcd1p degradation. (**E**) Positioned loops are gone upon Mcd1p-depletion. The bar chart shows the loop number called by the HiCCUPS program in *WT* (733) and *MCD1-AID* (5). (**F**) Genome-wide analysis confirms that the positioned loop signal is gone in *MCD1-AID* cells. Heatmaps were plotted with 200 bp resolution data for *WT* and *MCD1-AID* in ±5 kb regions surrounding the wild-type loop anchors.

The online version of this article includes the following figure supplement(s) for figure 2:

**Figure supplement 1.** Mitotic loops are cohesin-dependent.
**Figure supplement 2.** Comparison of chromosome contacts upon cohesin depletion of Micro-C XL versus Hi-C.

*Mitchison, 1994*; *Kimura and Hirano, 1997*; *Saitoh et al., 1994*; *Strunnikov et al., 1995*). Previous studies have shown that genome-wide loops are cohesin- not condensin-dependent (*Dauban et al., 2020*; *Schalbetter et al., 2017*). We tested whether the newly detected mitotic positioned loops in our study were also condensin-independent, by depleting Brn1p, a subunit of condensin (*Figure 2—figure supplement 1A*). The positioned loop signal was largely unaffected on the contact map (*Figure 2—figure supplement 1B*), as the number of detected positioned loops modestly decreased from 732 in wild-type to 585 when condensin was depleted (*Figure 2—figure supplement 1C*). Accordingly, the signal from the averaging analysis of the positioned loop regions was only slightly reduced when condensin was depleted (*Figure 2—figure supplement 1D*). In conclusion, cohesin is responsible for the formation of the vast majority of positioned loops in budding yeast mitotic chromosomes.

Contact density decay analysis showed a drop in contacts starting around the 10 kilobases region when cohesin was depleted (*Figure 2—figure supplement 1E*). This drop was absent when condensin was depleted, corroborating that cohesin, but not condensin was the major driver of loop formation. Furthermore, the number of the inter-chromosomal interactions was similar between wild-type and condensin-depleted cells, unlike cohesin depletion where these interactions doubled (*Figure 2—figure supplement 1F* and *Supplementary file 1*). Therefore, cohesin is also responsible for chromosome individualization and decreased inter-chromosome contacts in mitosis, as previously shown by Hi-C studies (*Schalbetter et al., 2017*).

To evaluate the effect of cohesin depletion on chromatin loops, we compared the changes in contact probability in Micro-C XL and previous Hi-C datasets (*Figure 2—figure supplement 2*; *Schalbetter et al., 2017*; *Dauban et al., 2020*). Contacts at a scale of ~10 kb were substantially decreased upon cohesin depletion in Micro-C XL and *Dauban et al., 2020* datasets, while the *Schalbetter et al., 2017* data shows a modest effect as the curves of contact probability and their derivative slopes are poorly separated from each other (*Figure 2—figure supplement 2A and B*). Previous studies have characterized that the contact enrichment at ~10–20 kb appears to be cohesin dependent. In addition to those contacts, the higher resolution Micro-C XL data revealed a finer and more pronounced separation between the curves of wild-type and cohesin depletion, showing that the cohesin-dependent structures overrepresent at the length of ~3–15 kb (*Figure 2—figure supplement 2C*). Our dataset presented a more pronounced decay in contact density upon cohesin inactivation (*Figure 2—figure supplement 2C*), suggesting that, in addition to the technical improvement on chromatin mapping, our degron-allele of Mcd1p might result in a better cohesin inactivation than the temperature-sensitive allele used in the other studies. In conclusion, the comparative analysis between Micro-C XL and Hi-C data highlights finer cohesin-dependent structures that were omitted in the previous studies, whose size-range is compatible with the range of positioned loops.

## Cohesin and condensin shape the rDNA locus in mitosis

The ribosomal DNA locus (RDN) has 75 to 100 tandem repeats of the 9.1 kilobases rDNA unit, with two transcribed loci (RDN37 and RDN5) spaced by two non-transcribed regions (NTS1 and NTS2). In interphase, the RDN locus forms a diffused puff adjacent to the rest of the chromosomal DNA, while in mitosis the RDN condenses into a thin line apart from the rest of the genome (*Figure 3A*; *Guacci et al., 1994*). Both cohesin and condensin complexes are needed for RDN condensation (*Guacci et al., 1997*; *Lavoie et al., 2000*; *Schalbetter et al., 2017*; *Lamothe et al., 2020*). However, their impact on chromosomal contacts within the RDN had not been reported.

We analyzed by ChIP-seq where cohesin (Mcd1p) and condensin (Brn1p) accumulate on one rDNA repeat (*Figure 3B*). When we plotted all the contacts in the RDN on one rDNA repeat, we detected a significant difference between interphase and mitosis (*Figure 3C*, first versus second panel). In interphase where the RDN is a decondensed puff, we detected widespread contacts all over the RDN that were caused by interactions within a repeat and/or between different repeats (intra- or inter-repeats are indistinguishable in sequencing data). While in mitosis where the RDN is a condensed line, the widespread off-diagonal signal was lost. Instead, we observed only the interactions within a repeat and/or between different repeats between the NTS regions where cohesin and condensin bind (*Figure 3B and C*; *Guacci et al., 1997*; *Lavoie et al., 2000*; *Lamothe et al., 2020*). Depletion of cohesin and/or condensin resulted in a contact map pattern that mimicked the wild-type in interphase, reflecting the lack of condensation of these mutants. We also analyzed the total number of contacts between the RDN and the rest of the genome and between the copies of the

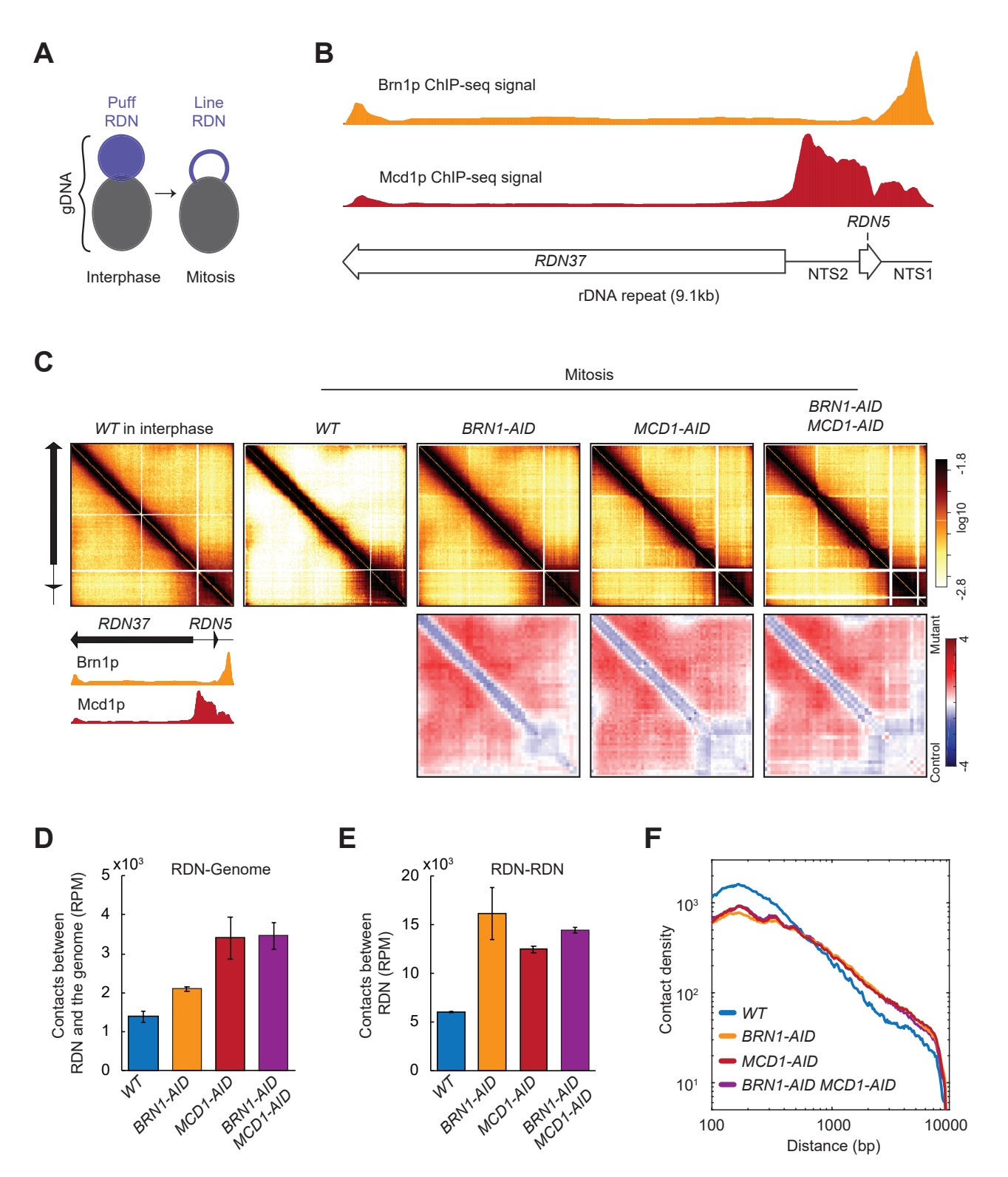

**Figure 3.** Ribosomal DNA (RDN) structure in mitosis is dependent on cohesin and condensin. (**A**) Model for RDN structure in interphase and mitosis. RDN (purple) is visualized as a dispersed puff separated from the rest of the genome (black). In mitosis, the RDN structures in a thin line. (**B**) Cohesin and Condensin accumulate in the 3' region of each rDNA repeat. ChIP-seq signal of condensin (Brn1p) in yellow shows how condensin accumulates on the NTS1 region. ChIP-seq cohesin (Mcd1p) in red shows cohesin accumulation on the NTS1 and 2. Below a cartoon for the genomic organization of

*Figure 3 continued*

the rDNA repeat, where each repeat is made of two transcribed regions (*RDN37* and *RDN5*) and two non-transcribed regions (NTS1 and NTS2). (**C**) Mitotic RDN structure resembles interphase upon cohesin or condensin depletion. Contact maps from one rDNA repeat from wild-type (*WT*) in interphase, followed by mitotically arrested cells in wild-type (*WT*), condensin Brn1p-depleted (*BRN1-AID*), cohesin Mcd1p-depleted (*MCD1-AID*), and cohesin- and condensin-depleted (*MCD1-AID BRN1-AID*) cells. Standard colormap scheme that uses the shades of red from white (no detectable interactions) to black (maximum interactions detected) in log10 scale. The contact fold-changes between mutants and wild-type were plotted using a diverging colormap with gaining contacts in mutants showing in red and losing contact in mutants showing in blue. Below the contact map, a cartoon for the genomic organization one RDN repeat and the ChIP-seq signal of condensin (Brn1p) in yellow and cohesin (Mcd1p) in red. (**D**) Cohesin and condensin depletion makes the rDNA less isolated from the rest of the genome. Normalized contacts between the rDNA and the rest of the genome in wild-type (*WT*), condensin Brn1p-depleted (*BRN1-AID*), cohesin Mcd1p-depleted (*MCD1-AID*). (**E**) Cohesin and condensin depletion increases the number of contacts between the RDN. Reads per million mapped reads (RPM) contacts between rDNA in wild-type (*WT*), condensin Brn1p-depleted (*BRN1-AID*), cohesin Mcd1p-depleted (*MCD1-AID*). (**F**) Cohesin and condensin depletion shows more long-range contacts on RDN. Interactions-versus-distance decaying curve shows the normalized contact density (y-axis) against the distance between the pair of crosslinked nucleosomes from 100 bp to 10000 bp (x-axis) for wild-type (*WT*), condensin Brn1p-depleted (*BRN1-AID*), cohesin Mcd1p-depleted (*MCD1-AID*), and condensin- and cohesin-depleted cells (*BRN1-AID MCD1-AID*).

rDNA. These contacts increased when cohesin and/or condensin were depleted from mitotically arrested cells (*Figure 3D and E*). Furthermore, both cohesin and condensin depletion similarly changed the contact decay pattern of the RDN from the wild-type (*Figure 3F*). In the mutants, we observed an increase in long-distance contacts, consistent with the presence of widespread off-diagonal interactions present in those mutants.

These results are consistent with the cytology of the RDN in interphase and mitosis. In interphase, the diffused RDN puff allows for many contacts within the RDN. The shared boundary of the RDN and the rest of the genome promotes contacts between them. In mitosis, the condensation of the RDN to a thin line away from the rest of the genome limits contacts both between RDN and the rest of the genome, and within the RDN. The ability of condensin and cohesin to restrict local contacts has not been reported previously. The restriction of local interactions could alter the accessibility of gene regulatory regions. This mechanism for cohesin and condensin regulation of gene expression is distinct from their proposed roles as regulators of long-range interactions between enhancers and promoters.

## Cohesin organizes chromosomal loops genome-wide

To better understand how the cohesin-rich regions organized the mitotic genome into positioned loops, we performed a more detailed comparison of the cohesin ChIP-seq and the Micro-C XL contact map. This analysis revealed three features of CARs in positioned loop organizations that were present throughout the yeast genome. First, each CAR formed at least two positioned loops, one by pairing with the CAR that precedes it on the chromosome, and one by pairing with the CAR that follows it (*Figure 4A*, Positioned loops from adjacent CARs). Thus, each CAR always exhibited bidirectional looping with the two adjacent CARs. Second, some CARs were also able to pair with CARs that were further down on the chromosome (+2, +3, etc.), resulting in larger positioned loops (*Figure 4A*, Positioned loops from distal CARs). We defined the positioned loops between distal CARs as loops expansion. Third, loop expansion often stopped at CARs with high Mcd1p ChIP-seq signal consistent with these CARs acting as barriers to loop expansion (*Figure 4A*, Barriers for loop expansion). Therefore, we draw a model that recapitulates how CARs organize the mitotic chromosomes into positioned loops through bidirectional looping, loop expansion, and barriers for loops expansion (*Figure 4B*).

We confirmed and elaborated on these observations through genome-wide analyses. The distribution of the distance between adjacent Mcd1p peaks overlapped with the distribution of the distance between two anchors of the called positioned loop, consistent with most positioned loops being called between adjacent CARs (*Figure 4C*). These positioned loops have a size distribution from around 3 kilobases to around 20 kilobases, with an average of around 8 kilobases. The size distribution of positioned loops falls within the range of chromosomal contacts previously detected in the contact decay analysis (*Figure 2—figure supplement 2*). We also analyzed loop expansion by quantifying the percentage of positioned loops being formed by proximal and distal CARs. Eighty-one percent of CARs form loops only with the adjacent CARs, while 19% formed positioned loops

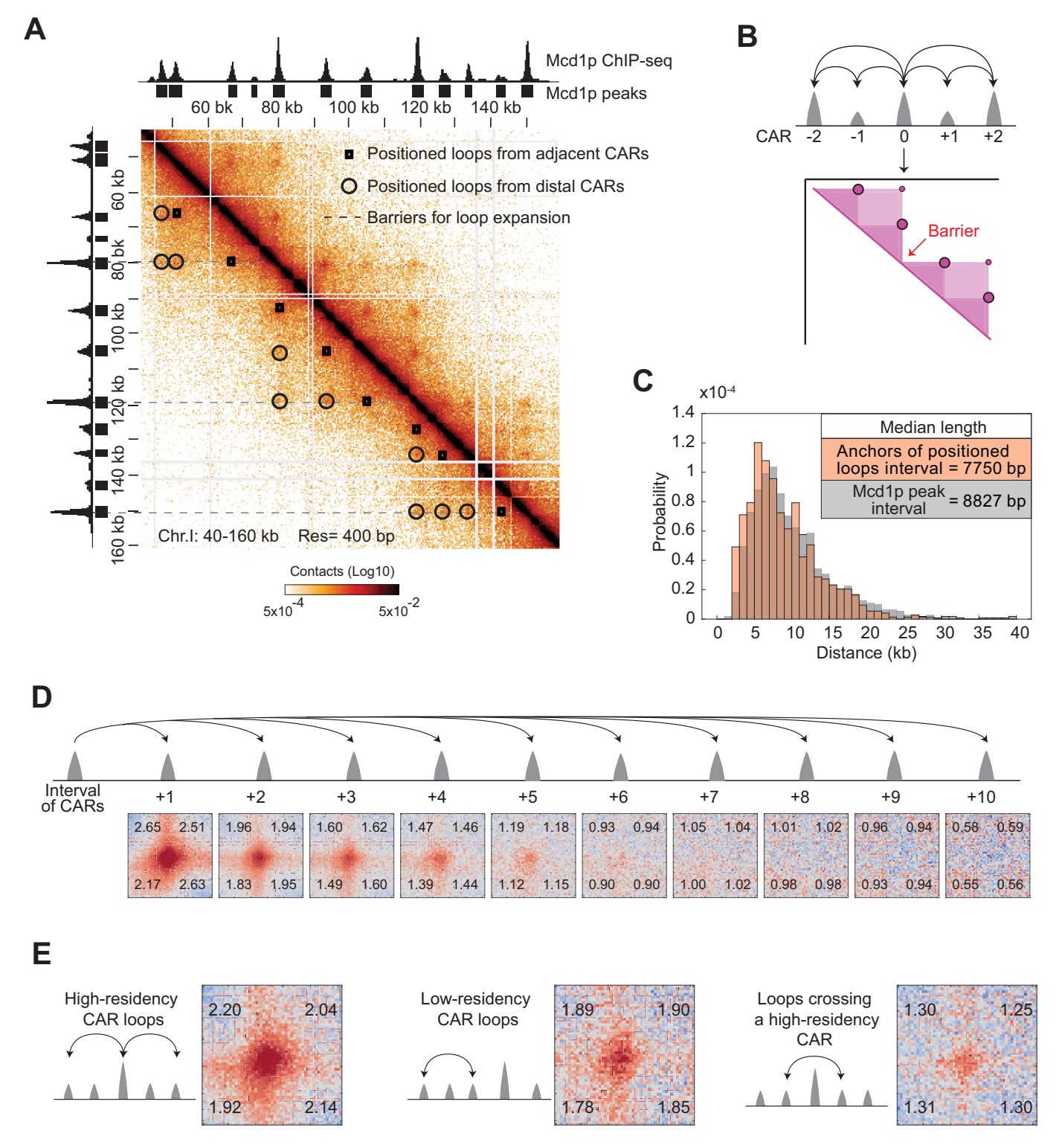

**Figure 4.** The organization of cohesin-dependent loops. (**A**) Features of chromatin loops organization from the contact map. Contact map showing the interaction in the 40–160 kb region of chromosome I overlay with the tracks for Mcd1p ChIP-seq signal and peaks. Squares indicate loops from adjacent CARs. Circles indicate loops from distal CARs. Dashed lines indicate the position of barriers that insulate from loop expansion. (**B**) Model for loops organization. Each CAR (Mcd1p peak) forms a loop with the CAR that follows and a loop with the CAR that precedes (Bidirectional loops from adjacent CARs). Some CAR can form loops with distal peaks (CAR0 with CAR+2, and CAR0 with CAR-2) (loop expansion). The high-residency CAR (peaks with

*Figure 4 continued on next page*

*Figure 4 continued*

the highest Mcd1p signal) are barriers to loop expansion. (**C**) The distribution of the size of positioned loops is correlated to the cohesin peak interval. The histogram shows the probability of the length distribution for positioned loop size and Mcd1p peak interval in the range from 0 to 40 kb. The inset table highlights the median length. (**D**) The positioned loop signal is detected till the +5 CARs interval genome-wide. Heatmaps were plotted with 200 bp resolution data for *WT*. A ±5 kb region surrounding each loop anchor and the corresponding CARs interval from +1 to +10. (**E**) The levels of cohesin binding in a loop anchor influence the strength of the corresponding loop and the ability to function as a barrier. Heatmaps were plotted with 200 bp resolution data for *WT*. We plotted a ±5 kb region either surrounding the high-residency CARs and the corresponding two-interval CARs (first panel); or surrounding the low-intensity CARs and the corresponding two-interval CARs (second panel); or surrounding a CAR and the corresponding two-interval CAR with a high-residency CAR in the middle (third panel).

The online version of this article includes the following figure supplement(s) for figure 4:

**Figure supplement 1.** Cohesin-dependent organization of loops.

with CARs further down the chromosome (*Figure 4—figure supplement 1A*). This analysis underestimated the number of CARs involved in loop expansion since the loop calling program performed poorly in detecting the spots that were far away from the diagonal (*Figure 4—figure supplement 1B*). To better estimate the extent of loop expansion and avoid loop calling bias, we piled-up the contact map for regions ranging from +1 to +10 CARs interval from each loop anchor in the genome. We detected a slow decrease in the loop signal as a function of the distance until the +5 CAR (*Figure 4D* quantified in *Figure 4—figure supplement 1C*). This data showed that the biggest positioned loops reached sizes of tens of kilobases. We conclude that loop expansion is a common feature of mitotic chromosomes and can occur over large distances.

We noted that the cohesin-rich regions in ChIP-seq presented a wide-range of peak heights. We ordered CARs based on the height of their peaks of cohesin binding from lower to higher, and we defined the top 10% as high-residency CARs. We also noted that the dots on the contact map presented different levels of intensity. We correlated the intensity of the dots/positioned loop in the contact map with the ChIP-seq peak height of the corresponding CAR (*Figure 4—figure supplement 1D and E*). We found that the intensity of the dots correlated with the intensity of the Mcd1p ChIP-seq signal on the corresponding CARs. We confirmed this correlation by comparing the genome-wide averaging signal from loops created by high- and low-intensity CARs (*Figure 4E*, first versus second panel). Moreover, we performed genome-wide averaging of the signal coming from the pairing of two CARs spaced by a high-residency CAR (*Figure 4E*, third panel). The signal detected was much weaker when compared to the signal between CARs that did not cross a high-residency CAR, suggesting that high-residency CARs block loop expansion. In conclusion, we described the key features of how cohesins organize the mitotic chromosomes into positioned loops: (1) each CAR always forms two positioned loops with its neighboring CARs in both directions; (2) looping can expand to distal CARs, up to ~5 CARs away in wild-type; (3) CARs with high-residency Mcd1p ChIP-seq signal function as a strong barrier to loop expansion.

## Chromosome loops form during S-phase

We showed that chromosomes from mitotically arrested cells were organized into positioned loops, but when these positioned loops formed during the cell cycle and how long they persist remained unanswered. To address these questions, we arrested cells in G1 with alpha-factor. We then released cells from the arrest and monitored loop formation and cohesin binding to chromosomes every 15 min for one entire cell cycle at two different temperatures 23°C and 30°C. We used two different temperatures to assess whether loop formation was affected by the rate of progression through the cell cycle. In budding yeast, the Mcd1p subunit of cohesin is synthesized at the beginning of S phase, allowing the assembly of a functional cohesin complex (*Guacci et al., 1997*; *Michaelis et al., 1997*). We detected the emergence of loop signal during DNA replication, concomitant with the synthesis and deposition of cohesin on chromosomes and its accumulation at CARs (*Figure 5A*, *Figure 5—figure supplement 1A and B*, *Ohno et al., 2019*). The signal for loops peaked in G2-M (around 75 min) and subsequently disappeared between mitosis and the next cell cycle (*Figure 5A*, *Figure 5—figure supplement 1A and B*). The timing of the disappearance of loops corresponded with the degradation of Mcd1p and inactivation of cohesin that normally occurs at the onset of anaphase (*Figure 5A*, *Figure 5—figure supplement 1A and B*; *Guacci et al., 1997*; *Michaelis et al., 1997*). Furthermore, previous studies in budding yeast showed that loop formation is independent

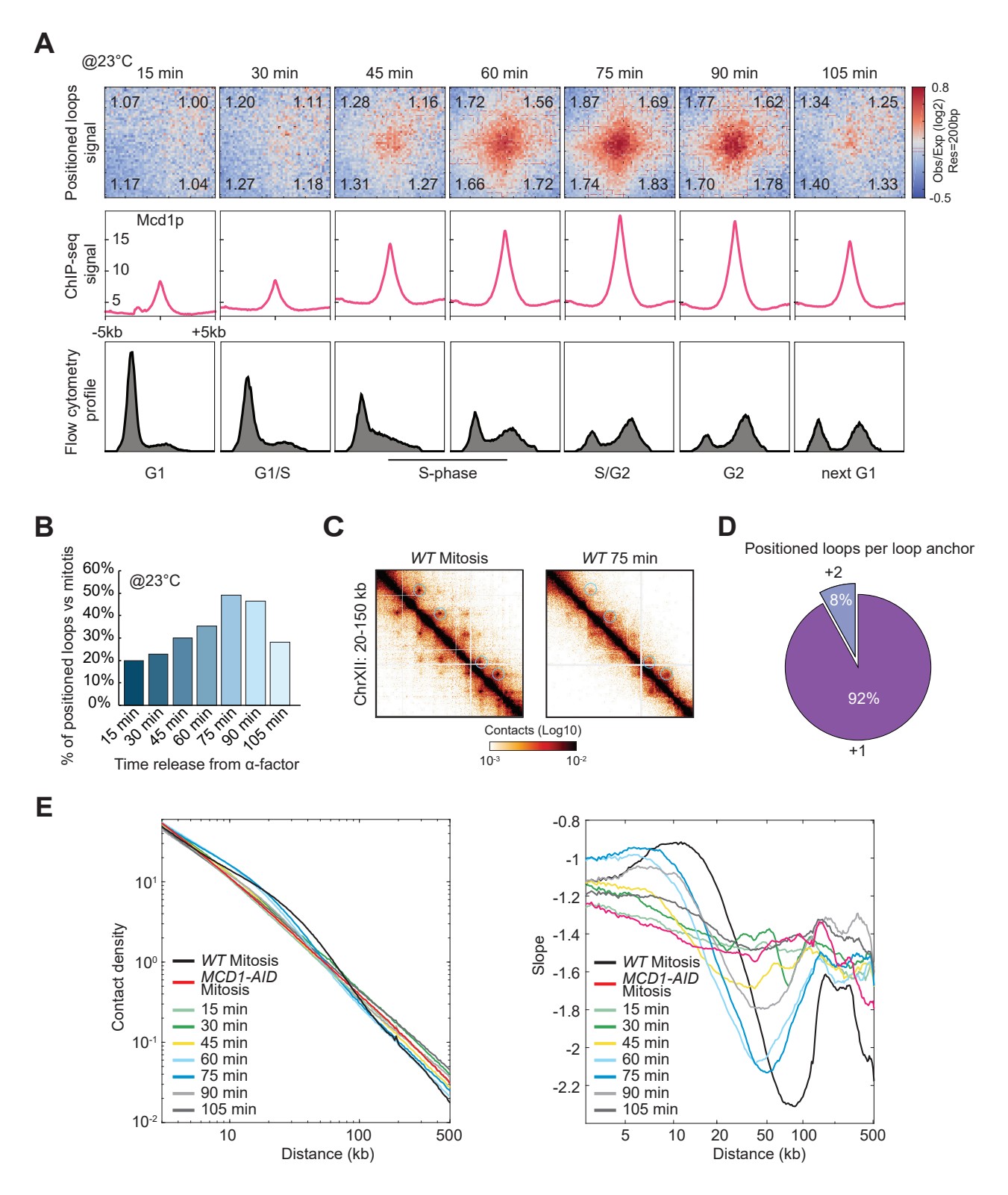

**Figure 5.** Chromatin loops are cell cycle dependent. (**A**) Chromatin positioned loops emerge upon cohesin deposition during DNA replication. The figure shows Micro-C (top), Mcd1p ChIP-seq (middle), and Flow cytometry profile (bottom) data across a time course after release from G1 arrest (15 to 105 min). Heatmaps were plotted with 200 bp data resolution across a time course in ±5 kb regions surrounding the loop anchors identified in the mitotically arrested wild-type data. Average Mcd1p peak intensity was plotted across ±5 kb regions around the center of loop anchors at each time

*Figure 5 continued on next page*

*Figure 5 continued*

point. Flow cytometry of 20,000 cells per time point. (B) Loop formation peaks in G2 (75 min time point after G1 release). Bar chart represents the percentage of positioned loop detected at each time point compared to the number of positioned loops found upon mitotic arrest. (C) Positioned loops detected during the time course are less defined on the contact map. Snapshots for cells arrested in mitosis (left) and after 75 min release from G1 (right) represent the chromatin interaction in the 20–150 kb region of chromosome XII. Note that snapshot for the 75 min time point is in a higher color contrast for a better loop visualization. (D) Only a small subset of loop anchors form positioned loops with distal anchors at the 75 min time point. The pie chart shows the percentage of loop anchors that form positioned loops either with the corresponding neighboring anchors (+1) or with distal anchors (+2) in the 75 min time point. (E) Cohesin-dependent chromosome contacts peak at the same time points where positioned loops were detected. Interactions-versus-distance decaying curve shows the normalized contact density (y-axis) against the distance between the pair of crosslinked nucleosomes from 100 bp to 500 kb (x-axis) for wild-type (*WT* arrested in mitosis), Mcd1p-depleted (*MCD1-AID* arrested in mitosis), and cells from each time point of the time course. On the right, the derivative slopes for each data were plotted against the distance from 3 to 500 kb.

The online version of this article includes the following figure supplement(s) for figure 5:

**Figure supplement 1.** Loop formation during the cell cycle.

of DNA replication and sister chromatin cohesion (*Dauban et al., 2020*; *Ohno et al., 2019*; *Schalbetter et al., 2017*). These results suggest that the cell cycle control of chromatin loops is achieved by the regulation of a functional cohesin complex and its deposition on chromosomes. The temporal binding of cohesin to chromosomes during the mammalian cell cycle also correlates with the presence of loops, suggesting a conserved mechanism (*Abramo et al., 2019*).

The number of positioned loops that were detected at different time intervals was smaller than the loops detected in mitotically arrested cells. (*Figure 5B*, *Figure 5—figure supplement 1C*). The 75 min sample had the most positioned loops but contained only 50% of the positioned loops observed in mitotically arrested cells. Also, in the 75 min sample, most of the spots were close to the diagonal as a result of the pairing of adjacent CARs (*Figure 5C and D*). This inability to detect loop expansion (+2, +3, etc.) was also evident for the other time points (*Figure 5—figure supplement 1A*). Pile-up analysis further confirmed the lack of loop expansion during the cell cycle. Only the +1 CAR interval showed substantial enrichment in the contact signal, while the enrichments are diminished in the distal CAR intervals. The contact decay analysis for cells ~60 to 75 min released from G1 had an enrichment in a similar range to the cohesin-dependent structures (or positioned loops) (*Figure 5E*). Interestingly, the contact enrichment in the cycling cells appeared to be slightly shorter than in the mitotically arrested wild-type cells, with the maximal signal peaking at ~8 kb. This decreased detection of positioned loops in the time course might reflect increased noise due to imperfect synchronization, an increase of cohesin binding to chromosome during the mitotic arrest, or the stabilization of loops and loop expansion during the mitotic arrest.

A key observation from our time course was that most of the cohesin deposition at CARs occurred 15 min before the appearance of the majority of positioned loop signal (*Figure 5A*, compare 45 vs 60 min). A model to explain the uncoupling of loop formation and cohesin binding to CARs is that this initial cohesin is not competent for loop extrusion. Loop extrusion may be mediated by a different pool of cohesin that is activated later in S phase.

## Cohesin regulators affect the size and location of chromosomal loops

Cohesin has many different activities that are regulated by multiple proteins. Two cohesin regulators, Wpl1p and Pds5p had been previously implicated in regulating chromatin loops in mammals and yeast (*Dauban et al., 2020*; *Haarhuis et al., 2017*; *Wutz et al., 2017*). In both yeast and mammals, Wpl1p functions as a negative regulator of cohesin, removing cohesin from chromosomes (*Chan et al., 2012*; *Kueng et al., 2006*). Counterintuitively, in budding yeast lacking Wpl1p, cohesin binding to CARs is reduced 50%, leading to reduced cohesion and condensation (*Guacci and Koshland, 2012*; *Lopez-Serra et al., 2013*; *Rolef Ben-Shahar et al., 2008*). This paradox can be explained if Wpl1p, by dissociating randomly bound cohesin from chromosomes, allows efficient accumulation of cohesin at CARs (*Bloom et al., 2018*; *Rolef Ben-Shahar et al., 2008*). Pds5p binds to cohesin and helps to maintain cohesin binding and cohesion from S-phase until the onset of anaphase (*Hartman et al., 2000*; *Panizza et al., 2000*). Pds5p also binds Wpl1p to facilitate the removal of a subset of cohesin (*Bloom et al., 2018*; *Rolef Ben-Shahar et al., 2008*). We wondered whether our high-resolution detection of loops might provide new mechanistic insights into the roles of Wpl1p and Pds5p in loop formation.

We depleted Wpl1p, arrested cells in mitosis (*Figure 6—figure supplement 1A*), and performed Micro-C XL to detect loop formation and Mcd1p ChIP-seq to detect the relative change in cohesin distribution on chromosomes. Depletion of Wpl1p reduced the amount of cohesin being detected at CARs (around half of the wild-type by visual inspection) as previously reported for yeast (*Rowland et al., 2009*; *Sutani et al., 2009*). It also caused a marked increase in the number of dots on the contact map (*Figure 6A and B*, *Figure 6—figure supplement 1B*). The number of called loops increased from 732 in wild-type to 1588 in cells depleted for Wpl1p, and visual inspection of the contact map showed many more dots linked by distal CARs. Since these positioned loops away from the diagonal were often missed by the loop calling program, the increase in the number of positioned loops was likely higher in Wpl1p-depleted cells. Therefore, Wpl1p restricts the size of positioned loops hence loop expansion in budding yeast as suggested previously from a Hi-C study (*Dauban et al., 2020*).

The additional positioned loops in Wpl1p-depleted cells appeared to result from the more distal pairing between existing loop anchors in wild-type cells, indicating an increase in loop expansion. Our genome-wide analyses confirmed that the loops present in wild-type were still present in Wpl1p-depleted cells (*Figure 6—figure supplement 1C*). Analysis of the contact decay curve showed that Wpl1p-depleted cells present an enrichment in a bigger size range than wild-type, confirming the presence of distal loops (*Figure 6—figure supplement 1D*). To assess the loop expansion quantitatively, we measured how many loops are formed by each loop anchor in one direction. The number of pairing partners for each loop anchor increased in Wpl1-depleted cells compared to wild-type (*Figure 6C*). We piled-up all the regions in the contact map ranging from +1 to +10 CARs interval from each positioned loop anchor in the genome (*Figure 6D* quantified in *Figure 6—figure supplement 1E*), to avoid loop calling bias. First, the signal from positioned loops to the close-distanced CARs (+1 and +2) was similar to wild-type, suggesting that Wpl1p-depleted cells retained the same group of positioned loops at these adjacent CARs. Second, Wpl1p-depleted cells presented an increased loop expansion capacity. In fact, while in wild-type cells loops with +5 CARs were weak, in Wpl1p-depleted cells the signal from loop anchors +5 was almost as robust as at +2. Moreover, in wild-type, the signal of loops extended to +5 CARs, while in Wpl1p-depleted cells it extended to +9 CARs. We conclude that Wpl1p restricts the number of positioned loops by decreasing the propensity and length of the loop expansion.

We depleted Pds5p from mitotically arrested cells (*Figure 6—figure supplement 1A*) and examined cohesin binding to chromosomes and loop formation. As published previously, Pds5p depletion caused a marked loss of cohesin all along the chromosome arms from visual inspection (*Figure 6E*, first column and 6G, first panel) (*Guacci et al., 2019*). Concomitantly with the loss of cohesin, the positioned loops disappeared from the chromosome arms genome-wide (*Figure 6E* first column, 6F, and *Figure 6—figure supplement 1F*). Thus, Pds5p is required for stable cohesin binding and formation of positioned loops on the arms of mitotic chromosomes.

We observed a single high-residency peak at the centromeres and a modest reduction of cohesin peaks in the pericentric regions (*Figure 6G*, second panel). These regions also exhibited an altered positioned loop distribution. The high-residency CAR at the centromere forms positioned loops with the other CARs along the chromosome, generating a prominent stripe pattern that can extend up to hundreds of kilobase (sometimes across an entire chromosome arm) (*Figure 6E*, second and third panels). Analysis of the contact decay curve showed that Pds5p-depleted cells lose the contact enrichment in the range of cohesin-dependent structures in wild-type, confirming the lack of genome-wide positioned loops (*Figure 6—figure supplement 1D*). Nevertheless, we detected an enrichment of the contacts in a much larger range, ~50–200 kb, and the loops stemming from the centromeres may contribute to the signal. These results suggest that in the absence of Pds5p, yeast cohesin can extrude chromatin from the centromere over long distances, forming loops of comparable size to the loops extruded by mammalian cohesin. Thus, Pds5p by stabilizing cohesin binding to chromosomes, it restricts loop expansion.

## Chromosome domains in mitosis

Another structural feature of the contact map was the presence of chromosomal interaction domains. We observed that the high-residency CARs demarcated the chromatin landscape into domains, presumably by blocking loop expansion (*Figure 7A*). These domains were found all over the chromosomes. Since domain calling is an empirical process in the contact map analysis, we

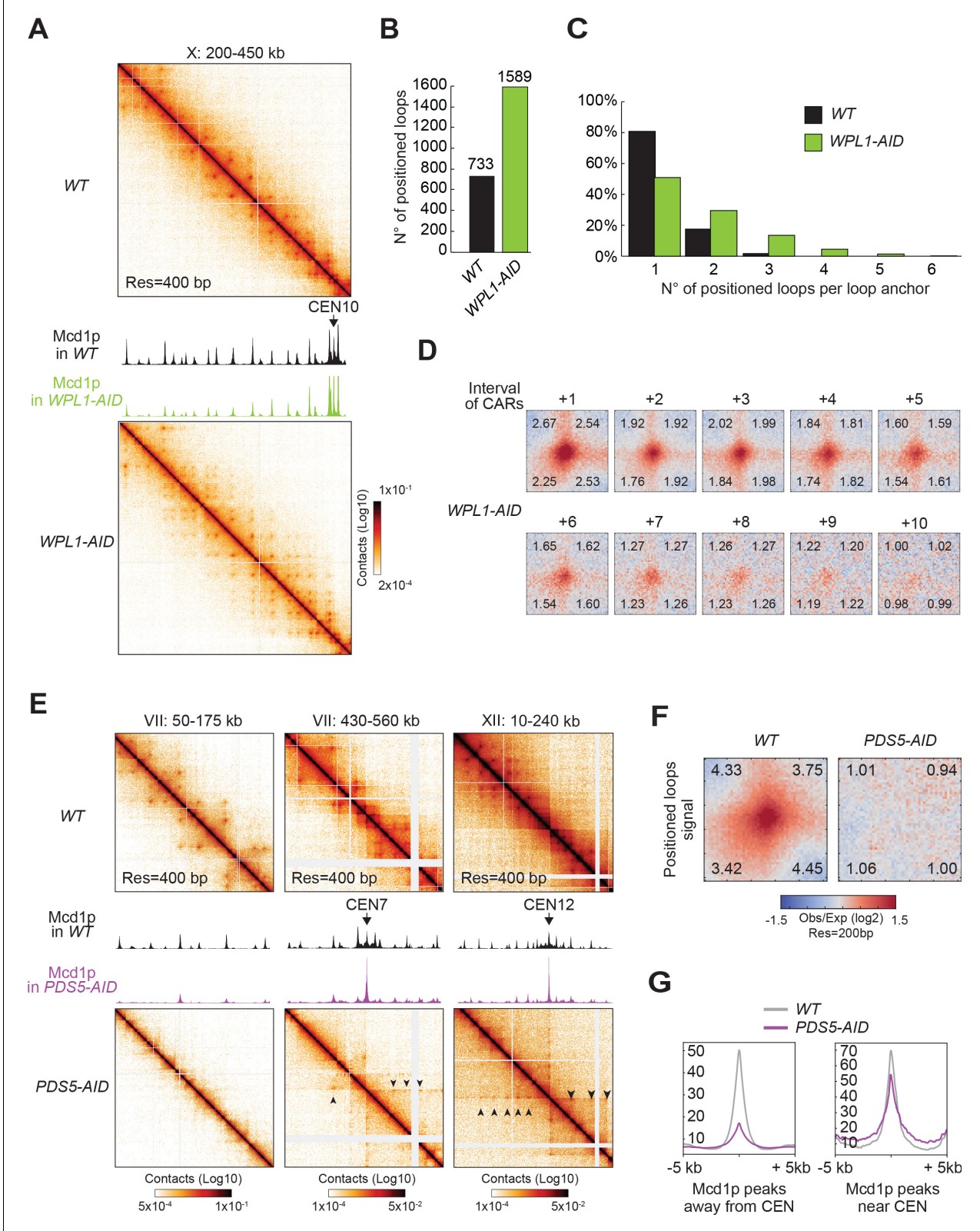

**Figure 6.** Wpl1p- and Pds5p-depletion perturb the looping pattern. (**A**) Wpl1p-depletion causes excessive loop expansion on the contact map. Contact maps for *WT* (top) and *WPL1-AID* (bottom) show the chromatin interactions over the 200–450 kb region of chromosome X. Mcd1p ChIP-seq data for *WT* and *WPL1-AID* are overlaid across the same region. The scale for ChIP-seq is 0–200. (**B**) Wpl1p-depletion doubles the number of positioned loops detected. Bar chart shows the total number of positioned loops called by HiCCUPS in *WT* and *WPL1-AID*. (**C**) Genome-wide analysis confirms loop

*Figure 6 continued on next page*

*Figure 6 continued*

expansion upon Wpl1p-depletion. Bar chart shows the percentage of the number of positioned loops per anchor for *WT* (black) and *WPL1-AID* (green). (D) Wpl1p-depletion results in the expansion of the loop signal to further distal loop anchors. Heatmaps were plotted with 200 bp data for *WT* and *WPL1-AID* in ±5 kb regions surrounding the wild-type loop anchors and the corresponding CARs interval from a +1 to a +10. (E) Pds5p-depletion results in positioned loops between each centromere and surrounding CARs. Contact maps for *WT* (top) and *PDS5-AID* (bottom) shows the chromatin interactions over a chromosome arm on the 50–175 kb region of chromosome VII (left), and the centromeric region at 430–560 kb on chromosome VII (middle), or another centromeric region at 10–240 kb of chromosome XII (right). Mcd1p ChIP-seq data for *WT* and *PDS5-AID* are overlaid across the corresponding region in each panel. The scale for ChIP-seq is 0–400. (F) Pds5p-depletion results in loss of signal at wild-type anchors for positioned loops. Heatmaps were plotted with 200 bp data resolution for *WT* and *PDS5-AID* in ±5 kb regions surrounding the wild-type positioned loop anchors. (G) Pds5p-depletion causes cohesin depletion from chromosome arms, but not from pericentric regions. Average Mcd1p peak intensity for *WT* and *PDS5-AID* were plotted over ±5 kb regions around the peak center. Peaks were sorted into two groups: (1) peaks on the chromosome arms (>30 kb away from the centromeres) (left); (2) pericentric peaks (<30 kb away from the centromeres) (right).

The online version of this article includes the following figure supplement(s) for figure 6:

**Figure supplement 1.** Wpl1p- and Pds5p-depletion alters the pattern of loops genome-wide.

extensively tested multiple methods to identify domains and found that the results from 'hicexplorer' match our visual inspection more consistently (*Wolff et al., 2018*). We applied this domain-calling algorithm at two kilobase resolution to the contact map of wild-type, Mcd1p-depleted, Brn1p-depleted cells arrested in mitosis, and published Micro-C XL on asynchronous cells (*Hsieh et al., 2015*; *Hsieh et al., 2016*; *Figure 7—figure supplement 1A*, and domain strength score is plotted in *Figure 7—figure supplement 1B and C*). Most of the domains detected in wild-type cells show boundaries that correlated with high-residency CARs. When cohesin was depleted, we observed a distinct profile of domain distribution. A visual inspection pointed out that some cohesin-depleted domains have a roughly similar boundary distribution to the ones detected in asynchronous cell populations (*Figure 7—figure supplement 1A*). We quantified this overlap by genome-wide analysis confirming that around 64% of the boundaries in the cohesin-depleted cells were located at the same positions as the boundaries detected in an asynchronous population, while only 14% overlapped with the boundaries of wild-type domains. The domains identified in the cohesin-depleted cells resembled the chromosomally-interacting domains (CIDs), which are typically delimited by promoters of highly transcribed genes, especially in a pair of divergently transcribed genes (*Hsieh et al., 2015*).

We further characterized the domains present in mitotic wild-type cells. To corroborate that the formation of these domains was dependent upon cohesin, we averaged the signal for each strain on the rescaled wild-type domains (*Figure 7B*). We confirmed that the strength of the wild-type domains was dependent on the presence of cohesin, but not condensin. Thus cohesin promotes domain formation as well as loops in budding yeast. To confirm that CARs were at the boundaries of the wild-type domains genome-wide, we plotted the cumulative Mcd1p ChIP-seq signal from wild-type cells at domains boundaries in wild-type, cohesin-depleted, and asynchronous cells (*Figure 7C* top). A marked enrichment of cohesin was present only at the wild-type domain boundaries confirming these domains are defined by CARs. Furthermore, cumulative curves for Mcd1p ChIP-seq signal showed that cohesin was enriched at wild-type domain boundaries when compared to cohesin-depleted, and asynchronous domain boundaries (*Figure 7C* bottom). We also observed a relocation of the domain boundaries from promoters (cohesin-depleted and asynchronous domains boundaries) to terminators (wild-type domains boundaries) (*Figure 7—figure supplement 1D*), as expected since the vast majority of CARs are located between the terminators of convergently transcribed genes (*Glynn et al., 2004*; *Lengronne et al., 2004*). Finally, the CAR domain boundaries presented a stronger Mcd1p ChIP-seq signal when compared to the other CARs genome-wide (*Figure 7D*), consistent with our finding that high-residency CARs had the capacity to insulate loop expansion to define domains. Our results suggest that chromosomes are organized into cohesin-dependent and cohesin-independent domains. The boundaries of the cohesin-dependent domains in wild-type mitotic cells are determined by CARs with strong cohesin binding, and we termed these new structures 'CAR domains'. These results are consistent with data from mammalian cells that show the presence of cohesin-dependent and cohesin-independent domains (referred to as compartmental domains). The compartmental domains are thought to be dependent on histones modifications and become more evident upon cohesin depletion (*Gassler et al., 2017*; *Rao et al., 2017*;

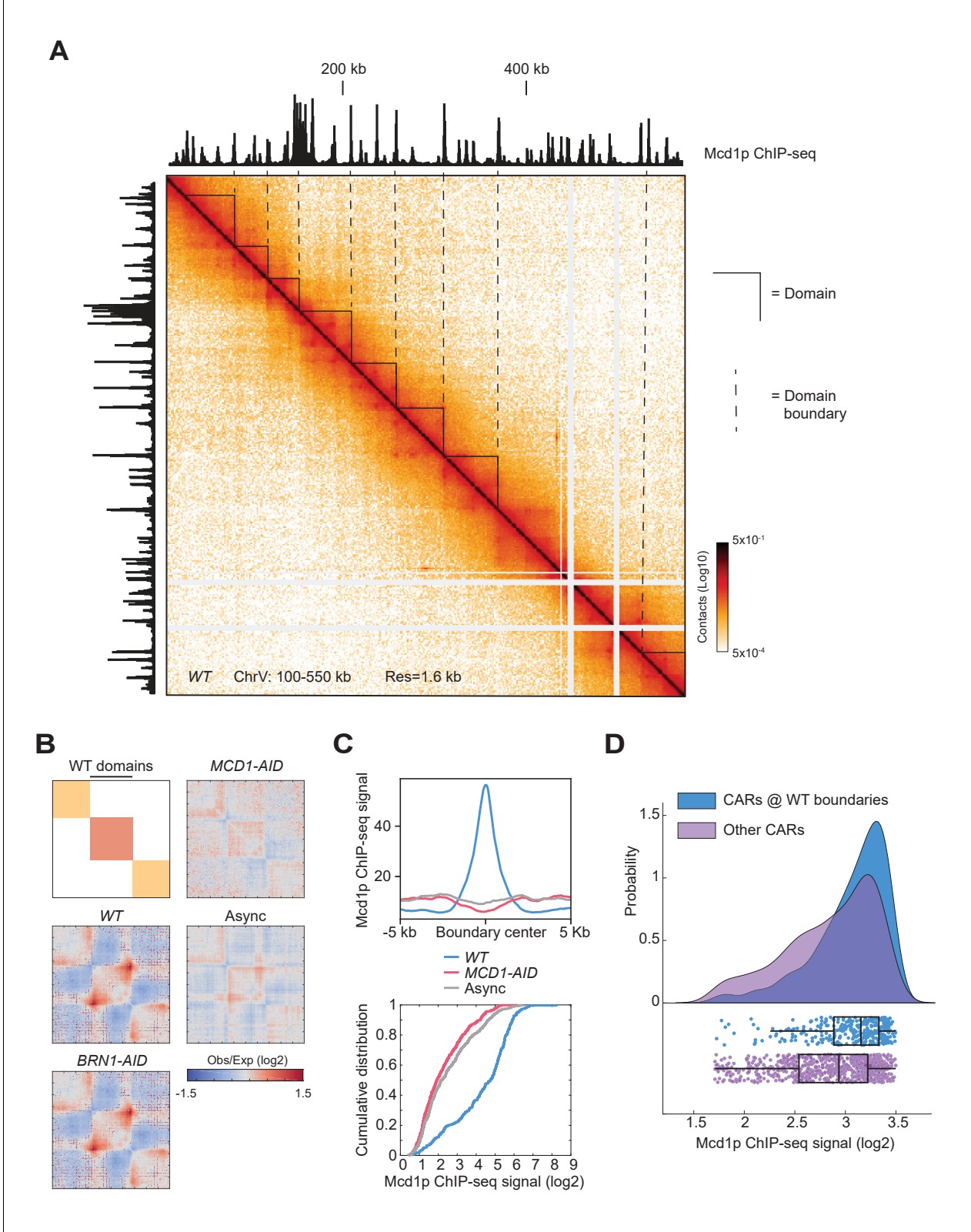

**Figure 7.** Cohesin depletion alters the domain landscape. (**A**) Contact map of mitotic chromatin in wild-type cells reveals the presence of CAR domains and their boundaries. Contact map showing the interaction in the 100–550 kb region of chromosome V overlays with the tracks for Mcd1p ChIP-seq signal. Triangles indicate the CAR domains' position. Dashed lines indicate the position of the CAR domain boundaries. (**B**) The signal composing CAR domains in wild-type is lost upon cohesin depletion. The wild-type (*WT*) domains are rescaled to the same length with left and right boundaries aligned

*Figure 7 continued on next page*

*Figure 7 continued*

on the plot (black bar). Genome-wide average domain/boundary strength for *WT, BRN1-AID, MCD1-AID*, and asynchronous cells were plotted as distance-normalized matrices over the CAR domains called in the wild-type (n = 306). (C) Cohesin is enriched at the boundaries of CAR domains in wild-type, but not at domains formed upon cohesin depletion. Mcd1p ChIP-seq data from wild-type cells were plotted separately over ±5 kb region (top) and cumulative curves were plotted separately as the function of log2 ratio of Mcd1p ChIP-seq signal (bottom) around the boundaries of CAR domains in wild-type (*WT*) (blue), the boundaries present upon Mcd1p-depletion (pink), or the boundaries present in asynchronous cells (grey). (D) CARs at wild-type domain boundaries have higher levels of cohesin binding than other CARs genome-wide. Cumulative curves show the probability distribution of Mcd1p ChIP-seq signal (log2) from CARs present at the wild-type CAR boundaries (*WT*) (blue), and from the other CARs (purple). Below the curves are plotted the corresponding Mcd1p ChIP-seq values of each CAR (blue dots for CARs at the CAR boundaries and purple dots for the other CARs). Box and whiskers plot indicates the median values and the quartiles distribution.

The online version of this article includes the following figure supplement(s) for figure 7:

**Figure supplement 1.** Cohesin depletion alters the chromatin domain landscape.

*Schwarzer et al., 2017*; *Wutz et al., 2017*). All together data suggest that cohesin-independent yeast CID are comparable to mammalian compartmental domains, while cohesin-dependent CAR domains in yeast are equivalent to the mammalian topologically associating domain (TAD).

## Discussion

In this study, we use Micro-C XL, a version of chromosome conformation capture with high-resolution to study chromosome architecture in mitotic yeast cells. This approach corroborated several findings from previous Hi-C studies including the presence of cohesin-dependent chromatin loops and the regulation of their size by cohesin regulators Wpl1p and Pds5p. However, our approach generated a robust signal for dots on the contact map, where loops with a defined position are distinct and easily detectable. We exploited this improved signal to reveal important new features about positioned loop distribution (size and genomic position) and insights into the molecular determinants of loop formation.

One feature of our study is the remarkable correlation between the anchors of positioned loops and cohesin associated regions (CARs). First, the genomic position of positioned loop anchors correlates almost perfectly with the position of CARs throughout the genome. The pairing of two anchors in a positioned loop always occurred between a CAR and the adjacent CAR on either side (bidirectional looping), but many CARs could also partner with more distal CARs (loop expansion). On average the portion between the two anchors for a positioned loop (~8 Kb) mirrors the average distance between adjacent CARs. Much larger loops were also observed (20–40 Kb), reflecting the distance between distal CARs. These data suggest that CARs determine the size and position of chromatin loops in mitotic budding yeast. Similarly, CARs have been suggested to determine the size and position of chromatin loops in meiosis (*Schalbetter et al., 2019*). The patterns of positioned loops in mitosis and meiosis are different, likely reflecting differences in the chromosomal structure and function in these two different yeast cell types.

The pervasive presence of CARs and their respective positioned loops along the chromosomes suggest that almost all, if not all, regions in the yeast genome can form a positioned loop. A similar omnipresence of positioned loops is also observed in yeast meiosis, and in mammalian cells (*Schalbetter et al., 2019*; *Hsieh et al., 2020*; *Krietenstein et al., 2020*; *Rao et al., 2014*). This shared global distribution suggests an important evolutionarily conserved function for looping chromatin, distinct from the previously proposed function in controlling the expression of a subset of genes. For example, the tethering of sister chromatids by cohesin is already known to promote DNA repair (*Sjögren and Nasmyth, 2001*; *Ström et al., 2004*; *Unal et al., 2004*). Looping by cohesin may contribute to the alignment of homologous repetitive elements between sister chromatids, thereby avoiding unequal sister chromatid exchange. Importantly, budding yeast provides an excellent system to test this and other potentially conserved physiological functions for cohesin-dependent chromatin loops.

The data from this and other studies in yeast revealed two differences between loop determinants in yeast and mammalian cells (*Schalbetter et al., 2019*; *Dauban et al., 2020*). First, in mammals, 86% of loop anchors depend on sites with CTCF and cohesin binding (*Rao et al., 2014*). In yeast cells loops formation and positioning depend on cohesin only, since yeast lacks CTCF or any

known DNA-binding factor that colocalizes with cohesin. Second, in budding yeast, each CAR forms loops with the adjacent CARs on either side (bidirectional), while in mammals loop formation is often directional correlating with the orientation of convergent CTCF sites (*Haarhuis et al., 2017*; *de Wit et al., 2015*; *Wutz et al., 2017*). The CTCF-independent mechanism in yeast may be a model for loop formation in most other eukaryotes that lack CTCF.

Indeed, from our study of loop anchors, we observed a positive correlation between the level of cohesin binding to a CAR and its probability of acting as a loop anchor in a positioned loop. A key to understanding this correlation comes from two observations. In yeast and mammals, cohesin binding to chromosomes is split between a stably bound pool and a pool that is more dynamic (*Gerlich et al., 2006*; *Hansen et al., 2017*). The ChIP-seq peaks at CARs result from stably bound cohesin (half-life >60 min *Eng et al., 2014*). The differences in ChIP-seq signal intensity at different CARs likely reflect the fraction of cells in the population with stably bound cohesin at those sites. Second, in our time-course experiment, the ChIP-seq signal from stably bound cohesin precedes the signal from positioned loops by 15 min.

Using these observations, we postulated a model to explain the correlation of the level of cohesin binding at CARs with the formation of loops and loop barriers (*Figure 8*). In S-phase, a functional cohesin complex is assembled and deposited on chromosomes and a pool of cohesin stably accumulates at CARs. In a cell population, some CARs are occupied by stably bound cohesin more often than others (High-residency CARs). After the binding at CARs, a pool of cohesin will start extruding the DNA forming a chromatin loop until it encounters two CARs that are stably occupied by cohesin. This encounter will stop the loop extrusion activity and the two CARs will be held together at the base of the loop, generating a positioned loop. Cohesin with loops extrusion activity could start looping bidirectionally in between the two occupied CARs, or cohesin from one occupied CAR can extrude DNA unidirectionally till it reaches the next occupied CAR. High-residency CARs are occupied by cohesin more often in a cell population, stopping incoming cohesin with loop extrusion activity efficiently and blocking its progression to CARs further down (barrier for loop spreading). CARs that are occupied less often in a cell population will stop the incoming looping cohesin in the few cells that have that CAR occupied, while the looping will proceed to CARs further down in the cells that have that CAR cohesin-free, resulting in bigger loops and loop spreading. Therefore, the frequency of a loop between any two CARs is determined in part by how often those CARs are occupied by cohesin, by the number of CARs between them, and by the number of incoming cohesin with loop extrusion activity. A similar model has been suggested for yeast meiotic chromosomes through polymer simulation modeling, and for mammalian mitotic chromosomes through an ingenious approach that allows differentiating contacts within and between sister chromatids (*Schalbetter et al., 2019*; *Mitter et al., 2020*). A key point will be to measure the percentage of yeast cells in a mitotic population that present a certain CAR occupied, and have the corresponding loop defined.

This model can also explain the signal of CAR domains and their boundaries. Our model postulates that positioned loops of different sizes being formed between a certain number of CARs. The sequences within each loop are more likely to interact with each other resulting in a 'loop domain' square signal on the contact map with the spot signal for the positioned loops at its vertices. The 'loop domains' square signal of different sizes coming from individual cells will be piled-up in the contact map, resulting in the observed CAR domain (*Figure 7*, right side). The high-residency CARs will block the loop extrusion activity functioning as a barrier, resulting in a boundary for the CAR domain. A similar model for loop extrusion and formation of domains was also postulated for mammalian cells (*Fudenberg et al., 2016*).

This model can also explain the pattern of positioned loops in Wpl1p- and Pds5p-depleted cells. Pds5p helps cohesin to establish cohesion but also forms a complex with Wpl1p to remove cohesin from chromosomes. It has been proposed that depletion of Wpl1p or Pds5p stabilizes the binding of dynamic cohesin, increasing the processivity of its looping activity and loop size (*Wutz et al., 2017*). However, this hypothesis alone could not explain additional differences between loop expansion and location in Wpl1p- and Pds5p-depletion revealed in this study.

In cells depleted for Wpl1p, the number of loops and loop expansion increased genome-wide compared to wild type. We can explain these changes in looping through our cohesin occupancy model and the fact that in budding yeast Wpl1p depletion reduced cohesin binding at CARs about 50% genome-wide (this study, [*Guacci et al., 2019*; *Rowland et al., 2009*; *Sutani et al., 2009*]) This

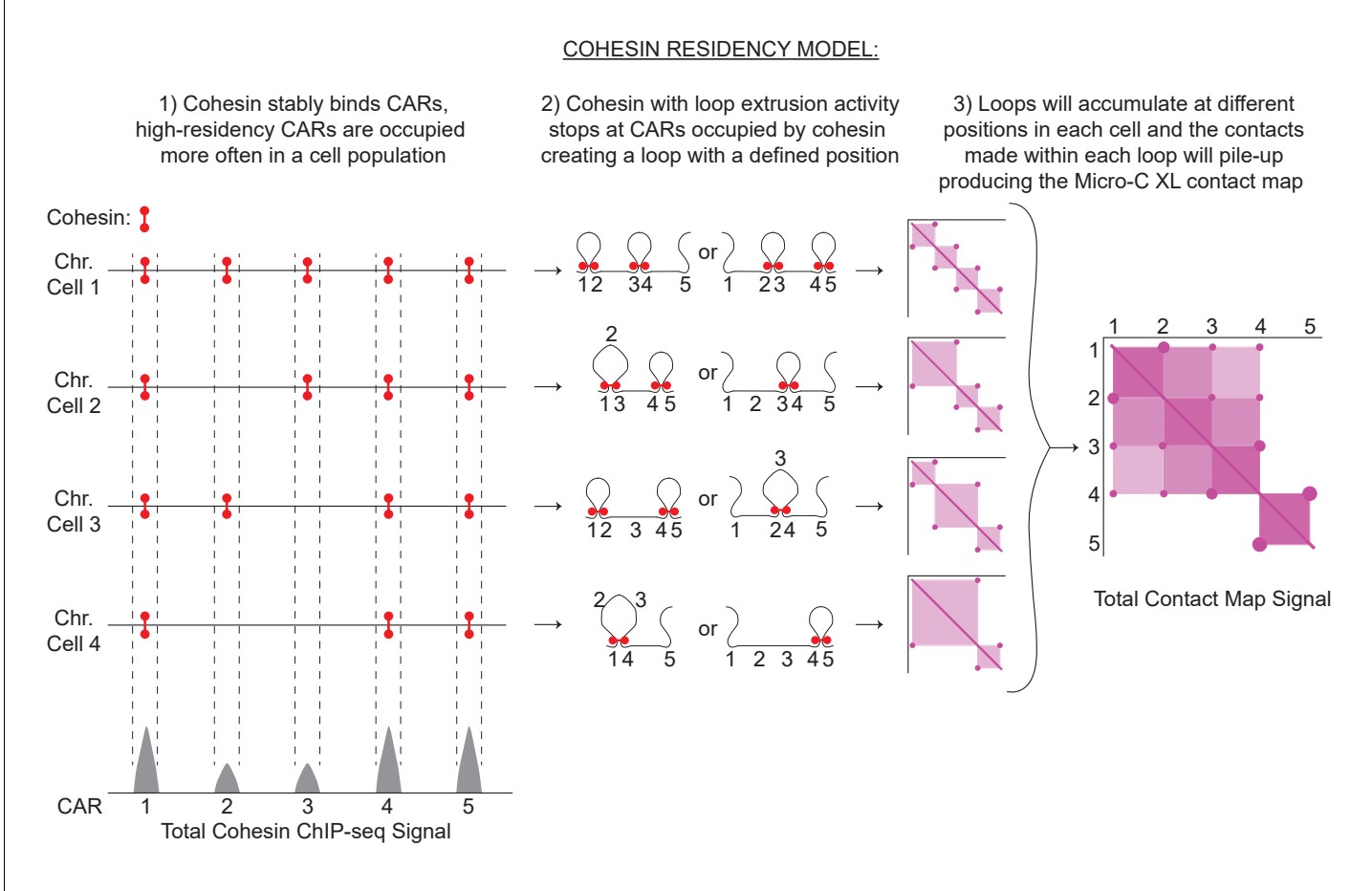

**Figure 8.** Cohesin residency model for loop formation. Loop-extrusion with a heterogenous cohesin residency gives rise to a heterogeneous pattern of chromatin looping and domains. The intensity of a cohesin ChIP-seq peak reflects the probability of the CAR being occupied by cohesin in a cell population. ChIP-seq peaks with high signals reflect CARs that are occupied with high probability by cohesin in a cell population. Low-signal peaks are CARs that are occupied by cohesin only in a fraction of cells. Cohesin with loop extrusion activity will start looping and stop when encountering two cohesin-bound CARs, creating a positioned loop. Every loop will result in a spot on the contact map from the stable interaction of the two CARs that function as loop anchors. Furthermore, the sequences inside the loop will have more probability of interacting with each other creating a loop domain visualized as a square in the contact map with spots at its vertices. Each spot and square signal from different positioned loops coming from individual cells will pile-up producing the final contact map signal detected by Micro-C XL with CAR domains with two high-intensity CAR as domain barriers and spots inside.

reduced binding of cohesin is not well understood but might reflect a role for Wpl1p in removing the non-specific binding of cohesin genome-wide to allow its accumulation at CARs (*Bloom et al., 2018*). We suggest that the defect in cohesin accumulation at CARs will result in a decrease in the number of CARs that are occupied by cohesin in the cell population, and will provide more dynamic cohesin with loop extrusion activity. As a result, in these cells increased amount of cohesin with loop extrusion activity can pair with occupied CARs that are more distant, allowing the formation of more loops and the expansion of loops size.

In Pds5-depleted cells, the number of loops decreased and more pronounced loop expansion (100's of kilobases) occurred only from the centromere compared to wild-type or Wpl1p-depleted cells. These differences can also be explained by our occupancy model and the dramatic differences in cohesin binding to different CARs in Pds5p-depleted cells. In ChIP-seq, we observed a prominent cohesin binding at the centromeres, greatly reduced binding in the centromere-proximal regions, and almost no binding at centromere-distal regions. We suggest that the high occupancy of cohesin at the centromere will always stop the looping of the dynamic cohesin. On the other side, the looping will proceed down the arm for long distances before being stopped by the occasional CAR site

occupied by stably bound cohesin. This process will result in a series of loops emanating from the centromeres to the arms in a cross pattern, as observed in our contact map. The low occupancy of the other CARs with stable cohesin is so low that dynamic cohesin is stopped rarely between these CARs, and therefore rarely generates loops on chromosome arms. Thus, the two very different looping patterns of Wpl1p- and Pds5p-depletion can be explained by our occupancy model. In summary, the patterns of cohesin binding to chromosomes and position of loops in wild-type and mutants suggest that loop expansion is restrained not only by decreasing the level of dynamic cohesin with looping activity but also by increasing the level of stably bound cohesin to CARs. In mammalian cells looping by dynamic cohesin may also be blocked by cohesin that is stably bound to chromosomes because of Sororin or CTCF.

The analysis of our results presents a novel interpretation of how yeast cells organize their genomes into loops. A previous study analyzed the chromosome contacts density produced with Hi-C using in silico modeling (*Schalbetter et al., 2017*). They predicted the presence of loops with an average size of 35 kb, randomly occurring on chromosomes, and covering around 35% of the mitotic yeast genome per cell. In contrast, our dataset detected chromosome contacts that start occurring at a size-range of 3–5 kb and peaks at 10–15 kb. Furthermore, we provided evidence that loops in yeast accumulate at defined positions throughout the genome, with a size range that is consistent with the aforementioned contact enrichment. We suggest that loops are much smaller than previously predicted, and they tend to accumulate at precise locations. The organization of the yeast genome into recurrent defined structures could potentially impact chromosome functions, much like the mammalian cells.

# Materials and methods

## Yeast strains and culture conditions

All strains used in this study are derived from the A364A background and their genotypes are listed in *Table 1*. YEP (yeast extract, peptone, and dextrose) medium was prepared as described previously (*Guacci et al., 1997*). For mitotically arrested cultures: cells were grown to early log phase (OD600 0.1–0.2) overnight at 23°C and arrested in G1 using the pheromone α factor at a concentration of 10 µM (Sigma Aldrich T6901-5MG) for 2.5 hr. Auxin was added to a final concentration of 500 µM to deplete AID-tagged proteins for 15 min (also to WT as control). Cells were released from G1 by washing with YPD containing auxin and 0.15 mg/mL pronase E (Sigma Aldrich) three times, and once with YPD plus auxin. After the last wash, cells were resuspended in YPD medium containing auxin and 15 µg/mL nocodazole (Sigma Aldrich) to arrest them in mitosis. Cultures were then once again incubated for 3 hr at 23°C. For the time course, WT cells were grown to early log phase (OD600 0.1–0.2) overnight at 23°C or 30°C and arrested in G1 using the pheromone α factor at a concentration of 10 µM (Sigma Aldrich T6901-5MG). Cells were released from G1 by washing with YPD with 0.15 mg/mL pronase E (Sigma Aldrich) three times, and once with YPD. After the last

**Table 1.** Strains.

| Strain name in the paper | Genotype | Request code | Reference |
|---|---|---|---|
| *WT* | MATa trp1Δ::pGPD1-TIR1-CaTRP1 lys4::LacO(DK)-NAT leu2-3,112 pHIS3-GFPLacI-HIS3:his3-11,15 ura3-52 bar1 | VG3620 | *Çamdere et al., 2015* |
| *MCD1-AID* | MATa MCD1-AID-KANMX6 ADH1-OsTIR1- URA3::ura3-52 lys4::LacO(DK)-NAT trp1-1 GFPLacI-HIS3: his3-11,15 bar1 leu2-3,112 | DK5542 | *Eng et al., 2014* |
| *BRN1-AID* | MATa BRN1-D375-3V5-AID2-HPHMX trp1Δ::pGPD1-TIR1-CaTRP1 lys4::LacO(DK)-NAT leu2-3,112 pHIS3-GFPLacI-HIS3:his3-11,15 ura3-52 bar1 | RL401 | *Lamothe et al., 2020* |
| *BRN1-AID MCD1-AID* | MATa MCD1-AID-KANMX6 BRN1-D375-3V5-AID2-HPHMX ADH1-OsTIR1- URA3::ura3-52 lys4::LacO (DK)-NAT trp1-1 GFPLacI-HIS3:his3-11,15 bar1 leu2-3,112 | RL406 | *Lamothe et al., 2020* |
| *PDS5-AID* | MATa PDS5-3V5-AID2-KANMX6 lys4::LacO(DK)-NAT pHIS3-GFP-LacIHIS3::his3-11,15 trp1-1 ura3-52 bar1 | TE228 | *Eng et al., 2014* |
| *WPL1-AID* | WPL1-3V5-AID-G418 TIR1-CaTRP1 bar1 LacO-NAT::lys4 leu2-3,112 GAL+ pHIS3-GFPLacI-HIS3:his3-11,15 ura3-52 | VG3629-3B | this work |

wash, cells were resuspended in YPD medium with no drugs and time points were collected every 15 min.

## Western blotting

To assess protein level, cells were collected, pelleted, washed with 1 × PBS and frozen at −80°C. Frozen pellets were resuspended in 200 µL of 20% TCA and broken for 40 s using a FastPrep-24 5G instrument (6.0 m/s each) (MP Bio). Lysates were diluted with 1 mL of 5%TCA and spun at 14,000 rpm for 10 min at 4° C. Pellets were resuspended in 2 × Laemmli buffer, boiled for 7 min, and spun. Cleared lysates were used for Western blotting on SDS-PAGE gels. Primary antibodies used were a mouse monoclonal anti-V5 used at a 1:5000 dilution (Invitrogen, Carlsbad, CA), mouse monoclonal anti-Tub1 used at 1:20,000 dilution, Anti-Mcd1p antibody (V. Guacci (via Covance)) at 1:10000 dilution, Secondary antibody used were an HRP-conjugated goat anti-mouse or rabbit at 1:25,000 (Bio-Rad, Hercules, CA).

## Flow cytometry

Fixed cells were washed twice in 50 mM sodium citrate (pH 7.2), then treated with RNase A (50 mM sodium citrate [pH 7.2]; 0.25 mg/ml RNase A; 1% Tween-20 [v/v]) overnight at 37°C. Proteinase K was then added to a final concentration of 0.2 mg/ml and samples were incubated at 50°C for 2 hr. Samples were sonicated for 30 s or until cells were adequately disaggregated. SYBR Green DNA I dye (Life Technologies, Carlsbad, CA) was then added at 1:20,000 dilution and samples were run on a Guava easyCyte flow cytometer (Millipore, Billerica, MA). 20,000 events were captured for each time point. Quantification was performed using FlowJo analysis software.

## Chromatin immunoprecipitation followed by sequencing (ChIP-seq)

Cells were grown and arrested at 23°C as described in the previous paragraph. ChIP-seq was performed as previously described (Costantino and Koshland, Mol Cell 2018). Briefly, 1 × 10$^{10}$ cells of exponentially growing cells (in 0.5 L of YEP+ 2% glucose) were fixed with paraformaldehyde (final 1%) for 1 hr. Glycine was added to stop the reaction at 0.125M for 10'. Cells were collected by centrifugation and washed once with PBS 1X buffer, once with TE 1X buffer and stored at −80C. Frozen cell pellets were resuspended in FA buffer (1% Triton X-100, 0.1% sodium deoxycholate,0.1% SDS, 50 mM HEPES, 150 mM NaCl, 1 mM EDTA) and disrupted three times using a FastPrep-24 5G instrument for 40 s (6.0 m/s each) (MP Bio). Chromatin was sheared 10 × 30 s on/30 s off with a Bioruptor Pico (Diagenode). Supernatant with chromatin extract was isolated after a 10 min centrifugation at 14,000 g at 4°C. Immunoprecipitation of DNA-protein was performed using 4 µL of the polyclonal rabbit anti-Mcd1p (Covance Biosciences) overnight at 4°C or anti-V5 for Brn1p-V5 (Invitrogen). Antibody-bound lysates were incubated with 120 µL Protein A dynabeads (Invitrogen 10001D) for 1 hr. Beads were then washed and the DNA eluted according to standard ChIP protocols: five successive washes with 1X FA, 1X FA with 0.5M NaCl, Lithium chloride detergent (0.25M LiCl, 0.5% NP-40, 0.5% Sodium deoxycholate, 1 mM EDTA, and 10 mM Tris-HCl), and finally twice with TE buffer. DNA was then eluted from the beads with 300 ml of Elution buffer (1% SDS, 0.1M NAHCO3). To ensure antibody is fully removed samples are treated with 1 ml proteinase K (20 mg/ml) for 1 hr at 42 while shaking at 1000 rpm, and the ssDNA was then isolated using QIAGEN PCR purification kit (28104 QIAGEN) and eluted in 20 ml low EDTA TE. The library was prepared using the Accel-NGS 1S Plus DNA Library Kit (Swift Bioscience) following the manufacturer protocol. Libraries were sequenced using Illumina Hiseq 4000. The sequencing files were aligned to the SacCer3 yeast genome using Bowtie2 tool and bigwig files were normalized for number of reads. Identified regions of enrichment were called using MACS, with default settings.

ChIP-seq data are accessible from the GEO repository accession number GSE151416.

## Yeast Micro-C assay

We used the Micro-C XL protocol reported in *Hsieh et al., 2016* with some minor modifications. Yeast cultures in the indicated conditions were immediately crosslinked with 3% formaldehyde for 15 min at 30°C, then quenched with 250 mM glycine at room temperature for 5 min. Crosslinked yeast cells were pelleted by centrifugation at 4000 rpm at 4°C for 5 min and washed once with deionized water. Pelleted yeast cells were then spheroplasted with 250 ug/mL Zymolyase

(MP08320932) in the Buffer Z (1M sorbitol, 50 mM Tris-HCl pH 7.4, 10 mM β-mercaptoethanol) at 30°C with vigorously shaking. Complete spheroplasting usually takes up to 40 to 60 min at which the milky solution turns to translucent, or the OD550 value drops to about half. Spheroplasts were washed once by 1X ice-cold PBS and then spun down at 4000 rpm at 4°C for 2 min. Pellets were resuspended in the DSG crosslinking solution (3 mM disuccinimidyl glutarate (ThermoFisher #20593) in 1X PBS) and incubated at 30°C for 40 min with gentle shaking. Crosslinking reaction was quenched with 400 mM glycine at room temperature for 5 min. Cells were pelleted by centrifugation at 4000 rpm at 4°C for 10 min and washed once with ice-cold PBS. Crosslinked cells can be stored at −80°C for few months or proceed to chromatin fragmentation immediately. We note that using freshly made formaldehyde and DSG solution is critical to generate high-quality Micro-C data.

Crosslinked cells were permeabilized with MB1 (50 mM NaCl, 10 mM Tris-HCl pH 7.5, 5 mM MgCl2, 1 mM CaCl2, 0.5 mM Spermidine, 1.43 mM β-mercaptoethanol, 0.005% NP-40) on ice for 5 min. Chromatin was fragmented into ~90% mononucleosome and ~10% dinucleosome with Micrococcal Nuclease (MNase) (Worthington Biochem #LS004798). We recommend titrating the MNase digesting condition for each sample. Typically, we treat cells with various pre-titrated MNase concentrations and then incubate them at 37°C for 10 min with vigorously mixing. MNase reaction was inactivated by adding 2.5 mM EGTA and incubated at 65°C for 10 min. Chromatin was pelleted by centrifugation at 16,000 g at 4°C for 5 min and washed twice with ice-cold MB2 (50 mM NaCl, 10 mM Tris-HCl pH 7.5, 10 mM MgCl2). We note that MNase titration to yield 90% monomer/10% dimers reduces un-digested (un-ligated) dimers contamination in Micro-C data.

MNase-digested fragments were then subject to the customized three-step end-repair protocol. First, chromatin was resuspended in the end-repair buffer (50 mM NaCl, 10 mM Tris-HCl pH 7.5, 10 mM MgCl2, 100 µg/mL BSA, 2 mM ATP, 5 mM DTT) and phosphorylated the 5'-ends with 25 units of T4 Polynucleotide Kinase (NEB #M0201) at 37°C for 15 min. Second, treating chromatin with 25 units of DNA Polymerase I, Large (Klenow) Fragment (NEB #M0210) in the absence of dNTPs at 37°C for 15 min allows the Klenow enzyme using its 3'-to-5' exonuclease activity to convert the mixed types of nucleosomal ends to the cohesive ends. Tight binding and chemical crosslinking between DNA and nucleosome are supposed to prevent the Klenow enzyme's 3'-to-5' exonuclease activity from chewing nucleosomal DNA all the way through. Third, Klenow enzyme's polymerase activity repairs the nucleosomal DNA ends to the blunt and ligatable ends upon the supplement of dNTPs including 66 µM dTTP, dGTP (NEB #N0446), biotin-dATP (Jena Bioscience #NU-835-BIO14), and biotin-dCTP (Jena Bioscience #NU-809-BIOX) in 1X T4 DNA ligase reaction buffer (NEB #B0202). The reaction was incubated at 25°C for 45 min with interval mixing and then inactivated with 30 mM EDTA at 65°C for 20 min. Chromatin was pelleted by centrifugation at 16,000 g at 4°C for 5 min and washed once with ice-cold MB3 (50 mM Tris-HCl pH 7.5, 10 mM MgCl2).

Biotinylated-nucleosomes were then subjected to the proximity ligation reaction with 5000 cohesive end units (CEU) of T4 DNA ligase (NEB #M0202) in 1X T4 DNA ligase reaction buffer (NEB #B0202) at room temperature for at least 1.5 hr with slowly shaking. After ligation, biotin-dNTPs at the DNA termini were removed with 500 units of Exonuclease III (NEB #M0206) in 1X NEBuffer 1 (NEB #B7001) at 37°C for 15 min. Samples were then reverse crosslinked with 1X proteinase K solution (500 ug/uL Proteinase K (ThermoFisher #AM2542), 1% SDS, 0.1 µg/µL RNaseA) at 65°C overnight. DNA was extracted by the standard Phenol:Chloroform:Isoamyl Alcohol (25:24:1) and ethanol precipitation procedure. DNA was then loaded and purified on ZymoClean column. Purified DNA was separated on a 3% NuSieve GTG agarose gel (Lonza #50081). The gel band corresponding to the size of dinucleosomal DNA (~250 to 350 bp) was cut and purified with Zymoclean Gel DNA Recovery Kit (Zymo #D4008). We note that the size-selection for DNA larger than 200 bp greatly reduces the ratio of unligated monomer in Micro-C data.

Micro-C sequencing library was generated by using NEBNext Ultra II DNA Library Prep Kit for Illumina (NEB #E7645) with some modifications. Purified DNA was end-repaired using End-it from Lucigen. 17.5 µl DNA was added to 2.5 ul 10X end-it buffer, 2.5 ul 10x ATP and 2.5 µl dNTP 10x, and finally 0.5 end-it enzyme was added. The mix was incubated at 25°C for 45 min and at 70°C fro 10 min. Then the Biotinylated-DNA was captured by Dynabeads MyOne Streptavidin C1 beads (ThermoFisher #65001) in 1X BW buffer (5 mM Tris-HCl pH 7.5, 0.5 mM EDTA, 1 M NaCl) at room temperature for 20 min with gentle rotation. DNA-conjugated beads were washed twice with 1X TBW buffer (0.1% Tween20, 5 mM Tris-HCl pH 7.5, 0.5 mM EDTA, 1 M NaCl) at 55°C for 5 min with vigorously shaking, rinsed once with Tris buffer (10 mM Tris-HCl pH 7.5), and then resuspended in Tris

buffer. We then performed 'on-bead' end-repair/A tailing and adapter ligation following the NEB protocol. Adapter-ligated DNA was washed once with 1X TBW and rinsed once with Tris buffer before PCR amplification. Micro-C library was generated by using KAPA HiFi HotStart ReadyMix (Roche #KK2601) or Q5 High-Fidelity 2X Master Mix (NEB #E7645) with commercial recommended conditions. We recommend using a minimal PCR cycle to reduce PCR duplicates, typically ranging from 8 to 12 cycles that can generate high-quality data for yeast samples. Prior to sequencing, purifying the library twice with 0.85X AMPure XP beads (Beckman #A63880) can eliminate primer dimers and adapters. We used Illumina 100 bp paired-end sequencing (PE100) to obtain ~50M reads for each replicate in this study.

Micro-C data are accessible from the GEO repository accession number GSE151553.

## Preprocessing Micro-C data from fastq to contact matrix

Each fastq mate was mapped to the yeast sacCer3 genome independently using Bowtie 2.3.0 (*Langmead and Salzberg, 2012*) with 'very-sensitive-local' mode. Aligned reads were paired by the read name. Pairs with multiple hits, low MAPQ, singleton, dangling end, self-circle, and PCR duplicates were discarded. Contacts shorter than 200 bp (unligated monomer) were also removed. Output files containing all valid pairs were used in downstream analyses. Valid Micro-C contact read pairs were obtained from the HiC-Pro analysis pipeline (*Servant et al., 2015*), and the detailed description and code can be found at https://github.com/nservant/HiC-Pro. We recommend checking these quality control statistics before moving forward to the downstream analysis: (1) bowtie mapping rate; (2) reads pairing percentage; (3) ratio of sequencing artifacts; (4) ratio of cis/trans contacts; (5) unligated monomer percentage. If any of the above statistics is not in the optimal range, one might consider checking mapping and filtering parameters or further optimize the Micro-C experiment.

Valid Micro-C contacts were assigned into the corresponding 'pseudo' nucleosome bin. The bin file was pre-generated from the yeast sacCer3 genome by the 100 bp window that virtually resembles the nucleosome resolution. The binned matrix can be stored in HDF5 format as COOL file format by using COOLER package (https://github.com/mirnylab/cooler; *Abdennur, 2017*) or HIC file format by using JUICER package (https://github.com/aidenlab/juicer; *Durand et al., 2016*). COOL and HIC are the standard 4DN formats that are compatible with many downstream analysis ecosystems (https://www.4dnucleome.org/). Contact matrices were then normalized by using iterative correction (IC) in COOL files (*Imakaev et al., 2012*) or Knight-Ruiz (KR) (*Knight and Ruiz, 2013*) in HIC files. Regions with low mappability and high noise were blocked before matrix normalization. We expect that matrix balancing normalization corrects systematic biases such as nucleosome occupancy, sequence uniqueness, GC content, or crosslinking effects (*Imakaev et al., 2012*). We notice that both normalization methods produce qualitatively equal contact maps. To visualize the contact matrices, we generated a compilation of COOL file with multiple resolutions (100 bp to 12,800 bp bins) that can be browsed on the Higlass 3D genome server (http://higlass.io; *Kerpedjiev et al., 2018*). In this study, all snapshots of Micro-C or Hi-C contact maps and the 1D browser tracks (e.g. ChIP-seq) were generated by the Higlass browser unless otherwise mentioned.

## Genome-wide contact decaying curve analysis

We only used intra-chromosomal contact pairs to calculate the contact probability in bins with exponentially increasing widths from 100 bp to 1 Mb. Contacts with a distance shorter than 100 bp were removed from the analysis to minimize noise introduced by self-ligation or undigested DNA products. Decaying curves in this study were normalized to the total number of contact pairs. The orientations of ligated DNA are parsed into 'IN-IN (+/-),' 'IN-OUT (+/+),' 'OUT-IN (-/-),' and 'OUT-OUT (-/+)' according to the readouts of Illumina sequencing (*Hsieh et al., 2015*). 'UNI' pairs are the combination of 'IN-OUT' and 'OUT-IN' because both orientations are theoretically interchangeable. We plotted the contact decaying curves with the 'UNI' orientation in this study unless otherwise mentioned.

Slopes of contact decay curves were obtained by measuring slopes in a fixed-width window. After the window searching through the entire range of curves, the slopes obtained from each window were connected and plotted against the genomic distance.

## Chromatin loop analysis

Loops (focal contact enrichment) in this study were identified by using the HICCUPS algorithm (*Rao et al., 2014*) in the JUICER package or Chromosight (*Matthey-Doret et al., 2020*). In HICCUPS analysis, loops were called with KR-normalized Micro-C contact matrices at 500 bp resolution, and filtered by a false discovery rate at 0.1 (calling options: -m 4096 k KR -r 500,500 f 0.1,0.1 p 6,8 -i 12,16 -d 2500,2500). In chromosight analysis, loops were detected with balanced contact matrices at 400 bp resolution and filtered by a cut-off in Pearson correlation coefficient 0.3 (calling options: `–pearson` 0.3 `–min`-dist 5000 `–max-dist` 1000000 `–min-separation` 1000 `–perc-undetected` 10). Hi-C data from *Schalbetter et al., 2017*; *Schalbetter et al., 2019*; *Paldi et al., 2020*; *Hsieh et al., 2016* were reanalyzed with the same loop calling option. Apparent false-positive hits (e.g. distance larger than 100 kb) were manually removed. We applied the lists of loop anchor to many downstream analyses by using Bedtools (*Quinlan and Hall, 2010*), including (1) comparison of loop anchors between Micro-C and Hi-C, (2) histogram of loop size, (3) cross-correlation with ChIP-seq data, and (4) the number of loops per anchor, etc. We noted that the current loop calling algorithm does not identify all the visible chromatin loops, mostly skipping a subset of distal and weaker loops.

Genome-wide loop intensity was assessed by the aggregate peak analysis (APA) (*Rao et al., 2014*). We used JUICER (*Durand et al., 2016*) or coolpuppy (*Flyamer et al., 2020*) packages to perform the analysis. In brief, loops were centered and piled up on a 10 kb x 10 kb matrix with 200 bp resolution data. The pile-up matrix was then normalized by KR, ICE, or distance, as indicated in the figure legend. Loops close to the diagonal were excluded or normalized by a random shift to avoid distance decay effects. The ratio of loop enrichment was calculated by dividing normalized center contacts in a searching window by the normalized corner submatrices.

We used the same approach to analyze the genome-wide target-centered loop intensity. Instead of aggregating at the intersection of loop anchors, the matrix is centered at the paired MCD1 ChIP-seq peaks. The pairs are selected for the adjacent peaks at distances shorter than 100 kb or the indicated peak intervals. The pile-up matrix was then normalized by KR, ICE, or distance, as indicated in the figure legend.

## Chromatin domain analysis

We used insulation score analysis (*Crane et al., 2015*) from the cooltools package (https://github.com/mirnylab/cooltools) (*Venev et al., 2020*) or the 'hicfindTADs' function from hicexplorer packages (https://github.com/deeptools/HiCExplorer) (*Wolff et al., 2020*) to identify sharp changes in chromatin interactions, which typically represent the domain boundaries. To identify kb-scale cohesin-mediated chromatin domains, we analyzed insulation profiles with Micro-C contact matrices at 2 kb resolution and 15 kb searching window. The sliding window scans across the entire genome and assigns the insulation intensity to its corresponding bin. The insulation scores were obtained and normalized as the log2 ratio of the individual score to the mean of the genome-wide averaged insulation score. Chromatin boundaries can be identified by finding the local minima along with the normalized insulation score. Boundaries overlapping with low mappability regions were removed from the downstream analysis.

For aggregate domain analysis (ADA), each domain was rescaled to a pseudo-size by $N_{i,j}=((C_i-D_{start})/(D_{end}-D_{start}), (C_j-D_{start})/(D_{end}-D_{start}))$, where $C_{i,j}$ is a pair of contact loci within domain D that is flanked by $D_{start}$ and $D_{end}$, and $N_{i,j}$ is a pair of the rescaled coordinates. The rescaled domains can be aggregated at the center of the plot with ICE or distance normalization.

## Additional information

### Funding

| Funder | Grant reference number | Author |
|---|---|---|
| National Institutes of Health | 1R35 GM-118189-01 | Douglas Koshland |

The funders had no role in study design, data collection and interpretation, or the decision to submit the work for publication.

## Author contributions
Lorenzo Costantino, Tsung-Han S Hsieh, Conceptualization, Data curation, Formal analysis, Investigation, Writing - original draft, Writing - review and editing; Rebecca Lamothe, Conceptualization, Investigation; Xavier Darzacq, Conceptualization, Funding acquisition; Douglas Koshland, Conceptualization, Funding acquisition, Writing - original draft, Writing - review and editing

## Author ORCIDs
Lorenzo Costantino (iD) https://orcid.org/0000-0002-4230-485X
Tsung-Han S Hsieh (iD) https://orcid.org/0000-0003-2094-0772
Xavier Darzacq (iD) http://orcid.org/0000-0003-2537-8395
Douglas Koshland (iD) https://orcid.org/0000-0003-3742-6294

## Decision letter and Author response
Decision letter https://doi.org/10.7554/eLife.59889.sa1
Author response https://doi.org/10.7554/eLife.59889.sa2

# Additional files

## Supplementary files
• Supplementary file 1. Summary of Micro-C and Hi-C data used in this study.
• Supplementary file 2. Summary of loop calling.
• Transparent reporting form

## Data availability
Sequencing data have been deposited in GEO under accession codes GSE151416 and GSE151553.

The following datasets were generated:

| Author(s) | Year | Dataset title | Dataset URL | Database and Identifier |
|---|---|---|---|---|
| Costantino L, Hsieh THS, Lamothe R, Darzacq X, Koshland D | 2020 | Cohesin residency determines chromatin loop patterns | https://www.ncbi.nlm.nih.gov/geo/query/acc.cgi?acc=GSE151416 | NCBI Gene Expression Omnibus, GSE151416 |
| Costantino L, Hsieh THS, Lamothe R, Darzacq X, Koshland D | 2020 | Cohesin residency determines chromatin loop patterns | https://www.ncbi.nlm.nih.gov/geo/query/acc.cgi?acc=GSE151553 | NCBI Gene Expression Omnibus, GSE151553 |

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
