## [Decision Letter]

**Acceptance summary:**

This study examines the structure of budding yeast chromosomes in mitosis using the high resolution chromosome capture technique, Micro-C XL. By analysis of specific mutants, high quality datasets are presented to unequivocally show that yeast chromosomes are organised into loops that are defined by cohesin. A model is presented for cohesin-dependent loop formation.

**Decision letter after peer review:**

Thank you for submitting your article "Cohesin residency determines chromatin loop patterns" for consideration by *eLife*. Your article has been reviewed by three peer reviewers, including Adèle L Marston as the Reviewing Editor and Reviewer #1, and the evaluation has been overseen by a Reviewing Editor and Jessica Tyler as the Senior Editor.

The reviewers have discussed the reviews with one another and the Reviewing Editor has drafted this decision to help you prepare a revised submission.

Summary:

The manuscript describes the generation and analysis of chromosome conformation changes in synchronised populations of budding yeast, *S. cerevisiae*, using a high-resolution method termed micro-C (an evolution of Hi-C) that was recently developed by one of the authors (Hsieh et al., 2015; Hsieh et al., 2016).

The authors describe a punctate interaction pattern that arises in metaphase-arrested cells, is dependent on the cohesin subunit Mcd1, and is altered upon depletion of the cohesin regulators Wpl1 and Pds5. The authors propose a modification of prior models of loop extrusion with barrier elements to interpret the microC patterns, specifically proposing that the stable binding of cohesin acts as a barrier to loop extrusion by dynamic cohesin.

The datasets themselves are informative, and will be of great use to the research community. It should be noted that some of the key findings were described in earlier work by Schalbetter et al., (2019) and (Dauban et al., 2020). The paper is technically impressive, the beautiful interaction patterns are compelling and the increased resolution of the micro-C both reinforces and extends conclusions from earlier work. However, there are a number of concerns related to how the analysis was done and how the findings are interpreted. The general perception of this manuscript would also be boosted by a small amount of additional analysis.

Essential revisions:

1) Though the data quality in the current study is unmatched, others have convincingly shown the presence of loops in budding yeast by Hi-C (Schalbetter et al., 2017, 2019; Dauban et al., 2020) and these studies need to be appropriately acknowledged throughout. Credit should be given to Dauban et al., 2020 for cohesin-dependent domains in G1, demonstrating cohesion-independent chromosome looping.

2) The authors must substantially rephrase and rewrite the text to avoid confusing the presence/absence of the punctate/grid-like focal interactions detected by Hi-C/micro-C with the presence/absence of "loops". The importance of this point cannot be overstated.

Critically, whilst it is correct to interpret focal matrix signals to represent the higher-order polymer interactions that can arise at loop bases, it is incorrect to infer that the lack of such focal interactions in a matrix indicates the lack of loops. Rather, the latter simply means that there are no "well-positioned" loops within the population of cells assayed.

Fundamentally, looped chromatin can be inferred from the relationship between the decay rate of interactions over distance (P(s)) without any explicit need to observe focal interactions-be this either because the dataset being studied lacks resolution to resolve such interactions (low-resolution HiC/binning, or low depth sequencing), or because loops may not arise at preferred locations (across the population). Indeed, prior mitotically-arrested yeast datasets were able to estimate both average loop length, and average loop number (per cell), despite the lower spatial resolution of the data analysed (Schalbetter et al., 2017).

3) A brief sentence subsection “Chromosome domains in mitosis” states that the boundaries of CAR domains are enriched at terminators (Figure 6—figure supplement 1C), where CARs are preferentially located. This point may be far from trivial. I indeed wonder whether the anchors of the loops presented in this paper overall correspond with terminators. A recent bioRxiv piece by the Cees Dekker and Uhlmann labs raises serious doubt as to whether yeast cohesin (in contrast to human cohesin) can in fact extrude loops. Yeast cohesin may still act as a topological anchor that when pushed along DNA enables the enlargement of a loop, but cohesin itself might not act as the motor. Of course the current manuscript does not need to fully address this point, but it would be relatively straightforward to assess whether the loop anchors genome-wide appear to correspond with terminators in general and/or with sites of convergent transcription in particular. Such an analysis could for sure add to the impact and the novelty of the current paper.

4) Do those CARs that act as boundaries then maybe correspond with convergent genes? If cohesin were pushed to these sites by transcription from two sides, this would also explain why these CARs have higher cohesin peaks by ChIP. It would be good to then also place these findings in the context of a recent paper by Paldi et al., on how convergent transcription shapes pericentromeres.

5) In Figure 2D, depletion of Mcd1 leads to the loss of focal loops. Yet, one can still observe domains. Also, here, it would be good to link these structures to the presence of genes. I would actually recommend that the authors throughout plot the genes above the matrices. A comparison of these domains with the Mcd1-independent domains observed in the previously published Micro-C work from asynchronous cells would also be useful here.

6) Figure 5—figure supplement 1: The plots in panels B and F suggest that loss of Wapl or Pds5 yields similar results, while e.g. panels A and E suggest that loss of these two factors yields very different results. Please explain.

7) The authors appear to conclude that it is non-random cohesin binding that defines the non-random location of loops (last sentence of Abstract for example). How do they exclude that it is not the non-random position of loops (generated by SMCs at their bases) that creates the non-random cohesin pattern?

8) Please specify in the text (for all experiments) for how long cells were arrested, and for the depletion experiments, how long after auxin was added were specific samples harvested? Have the authors confirmed depletion of AID-tagged proteins by western blotting in each experiment?

9) Critically, it is far from clear what genotypes and conditions are being compared. For example, is the wild type nocodazole sample also treated with auxin (as a control) when compared to the mcd1-AID, pds5-AID, and wpl1-AID mutants? Or are the AID strains processed {plus minus}auxin? Were durations of nocodazole arrest identical in the comparisons presented?

10) Please provide flow cytometry profiles to help demonstrate the homogeneity (or otherwise) of the samples/timepoints being analysed for each dataset in every figure. This is relevant for all data samples – especially when auxin may have been added for an unspecified time after the initial nocodazole arrest. Moreover, this is especially important for the S-phase timecourse presented in Figure 4. For these experiments, please explain why t = 0 minutes is not included as the starting state.

[Editors' note: further revisions were suggested prior to acceptance, as described below.]

Thank you for resubmitting your work entitled "Cohesin residency determines chromatin loop patterns" for further consideration by *eLife*. Your revised article has been evaluated by Jessica Tyler (Senior Editor) and a Reviewing Editor.

The manuscript has been improved but there are some remaining issues that need to be addressed before acceptance, as outlined below:

Although it is now clearer what had been shown in previous studies and how the current study advances the field, there are still instances where more context is required and there are some points that need revision to improve clarity.

Subsection “Micro-C XL reveals prevalent chromatin loops with a defined position in mitotic chromosomes”: The sentence should read "The pervasive presence of positions loops shown…."

Subsection “Micro-C XL reveals prevalent chromatin loops with a defined position in mitotic chromosomes”: "These positioned loops were mostly absent in contact maps using datasets from previously published Hi-C maps of mitotic cells (Figure 1C, bottom rows; Figure 1—figure supplement 1A)". This is not completely accurate – though there is no doubt that the Micro-C XL offers improved resolution, the positioned loops can be seen in the Schalbetter et al., data and are clearly present in the Paldi, 2020 data, just less distinct. This statement should be revised to "" These positioned loops were less obvious in contact maps using datasets from previously published Hi-C maps of mitotic cells (Figure 1C, bottom rows; Figure 1—figure supplement 1A)".

Subsection “Cohesin complex mediates loop formation”. As a general comment about the role of cohesin in loop formation in budding yeast, this should include a reference to Schalbetter, 2019. "To assess if cohesin was also needed for the formation of positioned loops in budding yeast…". The yeast meiotic observations (referenced in the Introduction) clearly demonstrate cohesin-dependence for loop positioning in yeast. To avoid confusion, it is suggested to write "whilst cohesin-dependent positioned loops have been clearly demonstrated in yeast meiosis, whether or not the same is true in mitotically-dividing yeast cells is unclear."

Subsection “Cohesin complex mediates loop formation”. These two paragraphs confirm conclusions already reported by others and this should be made clear. Schalbetter et al., 2017 already demonstrated that looping was mostly cohesin rather than condensin-dependent – this was the main conclusion of that study. Similarly, this point is confirmed in Daubain et al., 2020. Please preface these two paragraphs as confirming the previous findings in these studies, rather than presenting this as a new finding.

Subsection “Cohesin complex mediates loop formation”. A comparison to the meiotic result would provide better context and make it clear that this general concept has already been clearly demonstrated in the same organism albeit under different growth conditions. e.g. "Thus, we conclude, as in yeast meiosis (Schalbetter, 2019), that cohesin is required for positioned loop formation genome wide.".

Figure 2B. What are the units of the scale bar in B? How is this calculated? Is it a ratio? Is it a linear scale? Please specify in the legend.

Figure 2—figure supplement 2 and subsection “Cohesin complex mediates loop formation”. The main conclusion from this figure is that the increased resolution of the microC-XL allows the authors to draw the conclusion that the short distance interactions are also dependent on cohesin. Otherwise the datasets broadly similar and this point should be explicitly stated.

Subsection “Cohesin complex mediates loop formation”. It is not clear why this sentence was changed from the original. The revised text is now incorrect. Whilst it is true that cohesin is responsible for chromosome individualisation, cohesin activity leads to a *reduction* in interchromosomal contacts.

Subsection “Cohesin and condensin shape the rDNA locus in mitosis”: should be a single sentence "……(Guacci et al., 1994), while…" Also, please remove the comma within the text: "…loci (RDN37 and RDN5), spaced by two non-transcribed regions (NTS1 and NTS2)."

Subsection “Cohesin and condensin shape the rDNA locus in mitosis”: Schalbetter et al., 2017 also looked at the rDNA by Hi-C in the absence of cohesin and condensin with similar conclusions and should be cited here.

Subsection “Cohesin organizes chromosomal loops genome-wide”. For clarity, it is important to make it clear that the analysis of inter-anchor distances described here is only considering adjacent (+1) distances.

Subsection “Cohesin organizes chromosomal loops genome-wide” "While we detected…". As written, this sentence doesn't make sense grammatically. Please clarify what is meant.

Subsection “Cohesin organizes chromosomal loops genome-wide”. Without a direct test of causality, "confirming" is an overstatement. Please revise to "suggesting", or "supporting a model where.".

Subsection “Chromosome loops form during S-phase”. Please explain int the main text why two different temperatures were performed. The data are nice, but it is entirely unclear what conclusion the reader is expected to draw from the two-temperatures without further guidance.

Subsection “Chromosome loops form during S-phase”. "wild type cells" is stated, but do the authors mean 'wild-type cells arrested at mitosis with nocodazole'?

Subsection “Chromosome loops form during S-phase”. An alternative explanation is that there could be an increase in cohesin occupancy on chromosomes in mitotic arrested cells?

Subsection “Chromosome loops form during S-phase”: To make clear that this is speculation: "Loop extrusion may be mediated by a different pool of cohesin that is activated later in S phase.

Subsection “Cohesin regulators affect the size and location of chromosomal loops”. To improve readability, please add commas around the central clause as indicated here: "This paradox can be explained if Wpl1p, by dissociating randomly bound cohesin from chromosomes, allows efficient accumulation of cohesin at CARs (Bloom et al., 2018; Rolef Ben-Shahar et al., 2008)."

Subsection “Cohesin regulators affect the size and location of chromosomal loops”. What is the analysis that lead to the conclusion that the amount of cohesin detected at CARs was reduced 50%. The ChIP-seq data themselves are not calibrated. If analysis has not been done to determine this point, please revise the text accordingly. This also applies to the description for the data in pds5 mutant.

Subsection “Chromosome domains in mitosis”. To aid the reader, please provide a reference to Figure 7—figure supplement 1A here. However…the presented figure does not allow the reader to reach the conclusion that: "A visual inspection pointed out that most cohesin-depleted domains have a roughly similar boundary distribution to the ones detected in asynchronous cell populations." The few patterns that can be seen at this scale scale look to be quite different when comparing the mcd1-AID to the asynchronous Hsieh dataset.

Subsection “Chromosome domains in mitosis”. Where do these numbers come from? Is there a relevant graph that has been omitted?

Subsection “Chromosome domains in mitosis”. The authors should be wary of their conclusions drawn in this analysis. It is very likely that the domains detected by the algorithms in the wild type are almost entirely driven by the loop signals at the apex (and edge). (e.g. look at the patterns of signal enrichment in Figure 7B for wild type). Thus, obviously, if loops disappear, so will the ability to detect these domains bioinformatically.

Discussion section. Since this point has already clearly been demonstrated for preferred Rec8 sites in meiosis, Schalbetter, 2019 should be both mentioned and referenced here to avoid overstating the novelty of the findings.

Discussion section. Similarly, this finding was also demonstrated, clearly, by the meiotic yeast data, and thus should be described and cited.

Discussion section: As above, the conclusion that cohesin defines the loops is not entirely novel and previous work showing this should be cited here.

Discussion section. The model that is described is entirely congruent with the finding of the simulations employed by Schalbetter, 2019, which employed a stochastic model of loop expansion driven by extruders that could be blocked by barriers at preferred cohesin-binding sites (Rec8 in this case). As such, it would be entirely relevant to mention the congruence of the author's data (and model) with the findings of the polymer simulations developed by others.

---

## [Author Response]

Summary:The manuscript describes the generation and analysis of chromosome conformation changes in synchronised populations of budding yeast, *S. cerevisiae,* using a high-resolution method termed micro-C (an evolution of Hi-C) that was recently developed by one of the authors (Hsieh et al., 2015; Hsieh et al., 2016).The authors describe a punctate interaction pattern that arises in metaphase-arrested cells, is dependent on the cohesin subunit Mcd1, and is altered upon depletion of the cohesin regulators Wpl1 and Pds5. The authors propose a modification of prior models of loop extrusion with barrier elements to interpret the microC patterns, specifically proposing that the stable binding of cohesin acts as a barrier to loop extrusion by dynamic cohesin.The datasets themselves are informative, and will be of great use to the research community. It should be noted that some of the key findings were described in earlier work by Schalbetter et al., (2019) and (Dauban et al., 2020). The paper is technically impressive, the beautiful interaction patterns are compelling and the increased resolution of the micro-C both reinforces and extends conclusions from earlier work. However, there are a number of concerns related to how the analysis was done and how the findings are interpreted. The general perception of this manuscript would also be boosted by a small amount of additional analysis.Essential revisions:1) Though the data quality in the current study is unmatched, others have convincingly shown the presence of loops in budding yeast by Hi-C (Schalbetter et al., 2017, 2019; Dauban et al., 2020) and these studies need to be appropriately acknowledged throughout. Credit should be given to Dauban et al., 2020 for cohesin-dependent domains in G1, demonstrating cohesion-independent chromosome looping.

We cited Schalbetter et al., 2017, 2019; Dauban et al., 2020 as indicated. We would like to point out that the Dauban reference was already referenced in the text.

2) The authors must substantially rephrase and rewrite the text to avoid confusing the presence/absence of the punctate/grid-like focal interactions detected by Hi-C/micro-C with the presence/absence of "loops". The importance of this point cannot be overstated.Critically, whilst it is correct to interpret focal matrix signals to represent the higher-order polymer interactions that can arise at loop bases, it is incorrect to infer that the lack of such focal interactions in a matrix indicates the lack of loops. Rather, the latter simply means that there are no "well-positioned" loops within the population of cells assayed.Fundamentally, looped chromatin can be inferred from the relationship between the decay rate of interactions over distance (P(s)) without any explicit need to observe focal interactions-be this either because the dataset being studied lacks resolution to resolve such interactions (low-resolution HiC/binning, or low depth sequencing), or because loops may not arise at preferred locations (across the population). Indeed, prior mitotically-arrested yeast datasets were able to estimate both average loop length, and average loop number (per cell), despite the lower spatial resolution of the data analysed (Schalbetter et al., 2017).

We changed loops to “positioned loops” through the paper. To avoid any confusion to the scope of our paper, we defined the positioned loops initially and stated that we were going to focus only on those loops. We also indicated that random loops might be present and referenced the papers that analyzed those.

3) A brief sentence subsection “Chromosome domains in mitosis” states that the boundaries of CAR domains are enriched at terminators (Figure 6—figure supplement 1C), where CARs are preferentially located. This point may be far from trivial. I indeed wonder whether the anchors of the loops presented in this paper overall correspond with terminators. A recent bioRxiv piece by the Cees Dekker and Uhlmann labs raises serious doubt as to whether yeast cohesin (in contrast to human cohesin) can in fact extrude loops. Yeast cohesin may still act as a topological anchor that when pushed along DNA enables the enlargement of a loop, but cohesin itself might not act as the motor. Of course the current manuscript does not need to fully address this point, but it would be relatively straightforward to assess whether the loop anchors genome-wide appear to correspond with terminators in general and/or with sites of convergent transcription in particular. Such an analysis could for sure add to the impact and the novelty of the current paper.

Many papers demonstrated that the vast majority of CARs in yeast are preferentially located at terminators of convergent genes in budding yeast (80-85%), suggesting that cohesin is pushed to CAR sites by transcription (Lengronne, 2004, Glynn, 2004 and others). We now reference that in the text. However, transcription as a motor to push cohesin as a way to form positioned loops is extremely unlikely because a positioned loop contains multiple genes with different directions of transcription in between the two anchors.

4) Do those CARs that act as boundaries then maybe correspond with convergent genes? If cohesin were pushed to these sites by transcription from two sides, this would also explain why these CARs have higher cohesin peaks by ChIP. It would be good to then also place these findings in the context of a recent paper by Paldi et al., on how convergent transcription shapes pericentromeres.

Similar to the comment on point 3, being positioned at the terminators of convergent genes is not only a feature of the boundaries of CAR-domain (that correspond to high cohesin peak by ChIP-seq), but is a feature of almost all CARs in yeast.

5) In Figure 2D, depletion of Mcd1 leads to the loss of focal loops. Yet, one can still observe domains. Also, here, it would be good to link these structures to the presence of genes. I would actually recommend that the authors throughout plot the genes above the matrices. A comparison of these domains with the Mcd1-independent domains observed in the previously published Micro-C work from asynchronous cells would also be useful here.

We described in the text the presence of Cohesin dependent domain (CAR domain) and cohesin independent domain whose presence is revealed upon depletion of Mcd1p. This is similar to the human domains observed in the referenced papers. We have an analysis that shows the preferential location of the 2 different domain families with respect to the genes (CAR domains on terminators and cohesin-independent domains on promoters). We added the suggested comparison with previously published asynchronous cells and show that the majority of cohesin-independent domains overlap with the domains registered on asynchronous cells. In the revised manuscript, we extensively tested multiple methods (e.g., insulation scores, arrowhead, hicexplorer, and chromosight) to identify domains and found that the results from hicexplorer agree with our visual inspection more consistently. We updated the related figures and text with the new analysis. Given the packed distribution of genes and the size of chromosomes shown, a plot for genes is not really informative (it is really hard to pinpoint locations of genes features and orientations).

6) Figure 5—figure supplement 1: The plots in panels B and F suggest that loss of Wapl or Pds5 yields similar results, while e.g. panels A and E suggest that loss of these two factors yields very different results. Please explain.

The decay curves for *WPL1-AID* and *PDS5-AID* are significantly different. To make this difference more apparent to the reader, we plotted them together. The range/distance of slower decaying interactions reflect the range of size of the foci pattern on the contact map. *WPL1-AID* presents an enrichment of contacts slightly larger than WT, indicating an enrichment for bigger loops. PDS5-AID shows contact enrichment around 40 to 100kb and does not appear to have the enrichment in the same range as in the WT, suggesting the existence of extra wide-ranging loops is likely in the mutant (perhaps extending from centromere all the way to telomere).

7) The authors appear to conclude that it is non-random cohesin binding that defines the non-random location of loops (last sentence of Abstract for example). How do they exclude that it is not the non-random position of loops (generated by SMCs at their bases) that creates the non-random cohesin pattern?

In the time course experiment, the bulk of cohesin binding to CAR by ChIP-seq appears 15minutes before the bulk of “well positioned” loops. Hence our conclusion that cohesin binding defines the position of loops.

8) Please specify in the text (for all experiments) for how long cells were arrested, and for the depletion experiments, how long after auxin was added were specific samples harvested? Have the authors confirmed depletion of AID-tagged proteins by western blotting in each experiment?

We thank the reviewer for pointing this out. We had included this information in the original Material and methods section but it was unclearly presented. We added further description of the cultures and the strain used in the Material and methods section to avoid any confusion. Briefly, all the strains were treated with the same exact regimen. The same amount of Auxin was added (both in WT and AID strains) and treated with alpha-factor and nocodazole for the same period of time. In fact, most of the experiments were done on the same day side by side. The depletion was checked on every experiment, and we added the western blot to show that every AID tagged protein is not detected upon auxin. The strains were analyzed extensively in (Eng et al., 2014, Lamothe et al., 2020). We added the reference for the strains in the strain table.

9) Critically, it is far from clear what genotypes and conditions are being compared. For example, is the wild type nocodazole sample also treated with auxin (as a control) when compared to the mcd1-AID, pds5-AID, and wpl1-AID mutants? Or are the AID strains processed {plus minus}auxin? Were durations of nocodazole arrest identical in the comparisons presented?

Please refer to the comment on point 8. We clarify further the strains used in the material and methods. Basically, the strains: WT, MCD1-AID, BRN1-AID, MCD1-AID/BRN1-AID, WPL1-AID and PDS5-AID were treated with auxin and nocodazole for the same time.

10) Please provide flow cytometry profiles to help demonstrate the homogeneity (or otherwise) of the samples/timepoints being analysed for each dataset in every figure. This is relevant for all data samples – especially when auxin may have been added for an unspecified time after the initial nocodazole arrest. Moreover, this is especially important for the S-phase timecourse presented in Figure 4. For these experiments, please explain why t = 0 minutes is not included as the starting state.

All the mitotically arrested cells were treated with the same amount of drug (auxin and nocodazole) for the same length of time, as indicated in the original Materials and methods section. We checked their cell-cycle arrest by flow cytometry and all the strains were arrested in G2/M and are indistinguishable from one another. We agree that flow cytometry profiles are important in the time course and we added the flow cytometry profile below each panel.

Since cohesin-subunit Mcd1p is not present on alpha-factor arrested cells, but is needed to form loops (defined and undefined), we decided to start the analysis a few minutes after the release from alpha-factor. Furthermore, in our experience alpha-factor arrested cells (t=0) are slightly different from G1 cells. Therefore, we allowed the cells to recover from alpha-factor for a few minutes before beginning our analysis. We start seeing foci on the contact map at a later time point at both temperatures used, showing that we did not lose the initiation nor the range of loop formation.

[Editors' note: further revisions were suggested prior to acceptance, as described below.]

The manuscript has been improved but there are some remaining issues that need to be addressed before acceptance, as outlined below:Although it is now clearer what had been shown in previous studies and how the current study advances the field, there are still instances where more context is required and there are some points that need revision to improve clarity.Subsection “Micro-C XL reveals prevalent chromatin loops with a defined position in mitotic chromosomes”: The sentence should read "The pervasive presence of positions loops shown…."

We changed to “shown”.

Subsection “Micro-C XL reveals prevalent chromatin loops with a defined position in mitotic chromosomes”: "These positioned loops were mostly absent in contact maps using datasets from previously published Hi-C maps of mitotic cells (Figure 1C, bottom rows; Figure 1—figure supplement 1A)". This is not completely accurate – though there is no doubt that the Micro-C XL offers improved resolution, the positioned loops can be seen in the Schalbetter et al. data and are clearly present in the Paldi 2020 data, just less distinct. This statement should be revised to "" These positioned loops were less obvious in contact maps using datasets from previously published Hi-C maps of mitotic cells (Figure 1C, bottom rows; Figure 1—figure supplement 1A)".

We changed to: “These positioned loops were less frequent and less defined in contact maps using datasets from previously published Hi-C maps of mitotic cells.” This qualitative statement was subsequently validated by quantitative analysis that showed around 8 times more spots being called in our dataset.

Subsection “Cohesin complex mediates loop formation”. As a general comment about the role of cohesin in loop formation in budding yeast, this should include a reference to Schalbetter, 2019. "To assess if cohesin was also needed for the formation of positioned loops in budding yeast…". The yeast meiotic observations (referenced in the Introduction) clearly demonstrate cohesin-dependence for loop positioning in yeast. To avoid confusion, it is suggested to write "whilst cohesin-dependent positioned loops have been clearly demonstrated in yeast meiosis, whether or not the same is true in mitotically-dividing yeast cells is unclear."

In the Introduction we changed the sentence to underscore that cohesin-dependent loops in meiosis were previously detected in the referenced paper: “A recent study revealed a robust signal for cohesin-dependent positioned loops during meiosis (Schalbetter et al., 2019), revealing that yeast has indeed the capability to efficiently form positioned loops.”

In the Results section we focus our attention on mitotic yeast.

Subsection “Cohesin complex mediates loop formation”. These two paragraphs confirm conclusions already reported by others and this should be made clear. Schalbetter et al., 2017 already demonstrated that looping was mostly cohesin rather than condensin-dependent – this was the main conclusion of that study. Similarly, this point is confirmed in Daubain et al., 2020. Please preface these two paragraphs as confirming the previous findings in these studies, rather than presenting this as a new finding.

Subsection “Cohesin complex mediates loop formation” we clearly credited the mentioned papers: ‘In budding yeast, cohesin is needed for loops genome-wide (Dauban et al., 2020; Schalbetter et al., 2017).’

We added “Previous studies have shown that genome-wide loops are cohesin- not condensin-dependent (Dauban et al., 2020; Schalbetter et al., 2017). We tested whether the newly detected mitotic positioned loop in our study were also condensin-independent, by depleting Brn1p, a subunit of condensin (Figure 2—figure supplement 1A).”

Subsection “Cohesin complex mediates loop formation”. A comparison to the meiotic result would provide better context and make it clear that this general concept has already been clearly demonstrated in the same organism albeit under different growth conditions. e.g. "Thus, we conclude, as in yeast meiosis (Schalbetter, 2019), that cohesin is required for positioned loop formation genome wide.".

The Results section focuses on the mitotic yeast. The idea that chromosome loops in mitosis and meiosis should be similar because they come from the same organism under different growth conditions ignores the dramatic structural (especially in yeast) and functional differences between meiotic and meiosis chromosomes. Therefore, we think it is more appropriate to compare with the meiotic results in the introduction and discussion, as we now extensively do.

Figure 2B. What are the units of the scale bar in B? How is this calculated? Is it a ratio? Is it a linear scale? Please specify in the legend.

It is simply the number of reads from the ChIP-seq file. We specified that now in the legend.

Figure 2—figure supplement 2 and subsection “Cohesin complex mediates loop formation”. The main conclusion from this figure is that the increased resolution of the microC-XL allows the authors to draw the conclusion that the short distance interactions are also dependent on cohesin. Otherwise the datasets broadly similar and this point should be explicitly stated.

We are confused by this comment. We stated: “Previous studies have characterized that the contact enrichment at ~10 – 20kb appears to be cohesin dependent. In addition to those contacts, the higher resolution Micro-C XL data revealed a finer and more pronounced separation between the curves of wild-type and cohesin depletion, showing that the cohesin-dependent structures overrepresent at the length of ~3 – 15kb (Figure 2—figure supplement 2C).” This statement acknowledges the similarity in the big range and then focus on the differences between the datasets, as asked by the previous round of reviews.

Subsection “Cohesin complex mediates loop formation”. It is not clear why this sentence was changed from the original. The revised text is now incorrect. Whilst it is true that cohesin is responsible for chromosome individualisation, cohesin activity leads to a *reduction* in interchromosomal contacts.

We changed to: “Therefore, cohesin is also responsible for chromosome individualization and decreased inter-chromosome contacts in mitosis, as previously shown by Hi-C studies (Schalbetter et al., 2017).”

Subsection “Cohesin and condensin shape the rDNA locus in mitosis”: should be a single sentence "……(Guacci et al., 1994), while…" Also, please remove the comma within the text: "…loci (RDN37 and RDN5), spaced by two non-transcribed regions (NTS1 and NTS2)."

We changed to: “The ribosomal DNA locus (RDN) has 75 to 100 tandem repeats of the 9.1 kilobases rDNA unit, with two transcribed loci (*RDN37* and *RDN5*) spaced by two non-transcribed regions (NTS1 and NTS2). In interphase, the RDN locus forms a diffused puff adjacent to the rest of the chromosomal DNA, while in mitosis the RDN condenses into a thin line apart from the rest of the genome (Figure 3A) (Guacci et al., 1994).”

Subsection “Cohesin and condensin shape the rDNA locus in mitosis”: Schalbetter et al., 2017 also looked at the rDNA by Hi-C in the absence of cohesin and condensin with similar conclusions and should be cited here.

We included this reference now “Both cohesin and condensin complexes are needed for RDN condensation (Guacci et al., 1997; Lavoie et al., 2000; Schalbetter et al. 2017; Lamothe et al., 2020). However, their impact on chromosomal contacts within the RDN had not been reported.”

Subsection “Cohesin organizes chromosomal loops genome-wide”. For clarity, it is important to make it clear that the analysis of inter-anchor distances described here is only considering adjacent (+1) distances.

The inter-anchor distances described here is for every single positioned loop being called by the loop calling program, not just the +1.

Subsection “Cohesin organizes chromosomal loops genome-wide” "While we detected…". As written, this sentence doesn't make sense grammatically. Please clarify what is meant.

We changed to: “We detected a slow decrease in the loop signal as a function of the distance till the +5 CAR (Figure 4D quantified in Figure 4—figure supplement 1C). This data showed that the biggest positioned loops reached sizes of tens of kilobases.”

Subsection “Cohesin organizes chromosomal loops genome-wide”. Without a direct test of causality, "confirming" is an overstatement. Please revise to "suggesting", or "supporting a model where.".

We changed to “suggesting”.

Subsection “Chromosome loops form during S-phase”. Please explain int the main text why two different temperatures were performed. The data are nice, but it is entirely unclear what conclusion the reader is expected to draw from the two-temperatures without further guidance.

We added: We used two different temperatures to assess whether loop formation was affected by the rate of progression through the cell cycle.

Subsection “Chromosome loops form during S-phase”. "wild type cells" is stated, but do the authors mean 'wild-type cells arrested at mitosis with nocodazole'?

We changed to: “Interestingly, the contact enrichment in the cycling cells appeared to be slightly shorter than in the mitotically arrested wild-type cells.”

Subsection “Chromosome loops form during S-phase”. An alternative explanation is that there could be an increase in cohesin occupancy on chromosomes in mitotic arrested cells?

We changed to: “This decreased detection of positioned loops in the time course might reflect increased noise due to imperfect synchronization, an increase of cohesin binding to chromosome during the mitotic arrest, or the stabilization of loops and loop expansion during the mitotic arrest.”

Subsection “Chromosome loops form during S-phase”: To make clear that this is speculation: "Loop extrusion may be mediated by a different pool of cohesin that is activated later in S phase.

We changed as suggested.

Subsection “Cohesin regulators affect the size and location of chromosomal loops”. To improve readability, please add commas around the central clause as indicated here: "This paradox can be explained if Wpl1p, by dissociating randomly bound cohesin from chromosomes, allows efficient accumulation of cohesin at CARs (Bloom et al., 2018; Rolef Ben-Shahar et al., 2008)."

We changed as suggested.

Subsection “Cohesin regulators affect the size and location of chromosomal loops”. What is the analysis that lead to the conclusion that the amount of cohesin detected at CARs was reduced 50%. The ChIP-seq data themselves are not calibrated. If analysis has not been done to determine this point, please revise the text accordingly. This also applies to the description for the data in pds5 mutant.

We changed the text to explicitly indicate to the reader that the drop in signal in the 2 strains was assessed by simple visual inspection. The reduction presented in our dataset, reflects the magnitude of cohesin binding drop observed previously in other studies (referenced in the text).

We changed that to: “Depletion of Wpl1p reduced the amount of cohesin being detected at CARs (around half of the wild-type from visual inspection) as previously reported for yeast (Rowland et al., 2009; Sutani et al., 2009).”

“As published previously, Pds5p depletion caused a marked loss of cohesin all along the chromosome arms from visual inspection (Figure 6E, first column and 6G, first panel) (Guacci et al., 2019).”

Subsection “Chromosome domains in mitosis”. To aid the reader, please provide a reference to Figure 7—figure supplement 1A here. However…the presented figure does not allow the reader to reach the conclusion that: "A visual inspection pointed out that most cohesin-depleted domains have a roughly similar boundary distribution to the ones detected in asynchronous cell populations." The few patterns that can be seen at this scale scale look to be quite different when comparing the mcd1-AID to the asynchronous Hsieh dataset.

We added the figure reference. We changed the text to reflect that (some instead of most): A visual inspection pointed out that some cohesin-depleted domains have a roughly similar boundary distribution to the ones detected in asynchronous cell populations (Figure 7—figure supplement 1A). We quantified this overlap by genome-wide analysis confirming that around 64% of the boundaries in the cohesin-depleted cells were located at the same positions as the boundaries detected in an asynchronous population, while only 14% overlapped with the boundaries of wild-type domains.

Subsection “Chromosome domains in mitosis”. Where do these numbers come from? Is there a relevant graph that has been omitted?

We presented the overlap between the datasets just in the text, since it is just 2 numbers.

Subsection “Chromosome domains in mitosis”. The authors should be wary of their conclusions drawn in this analysis. It is very likely that the domains detected by the algorithms in the wild type are almost entirely driven by the loop signals at the apex (and edge). (e.g. look at the patterns of signal enrichment in Figure 7B for wild type). Thus, obviously, if loops disappear, so will the ability to detect these domains bioinformatically.

We used a sliding window algorithm to detect the boundary-delimited domains. In our test, the method is minimally impacted by the present of spots at the apex. Over 90% of boundaries/domains were recovered in the data after removing the focal signals at the corner spots. Furthermore, the majority of the domains are called by the presence of a distinct square signal that does not present a strong spot signal at their apexes.

Discussion section. Since this point has already clearly been demonstrated for preferred Rec8 sites in meiosis, Schalbetter, 2019 should be both mentioned and referenced here to avoid overstating the novelty of the findings.

We added: These data suggest that CARs determine the size and position of chromatin loops in mitotic budding yeast. Similarly, CARs have been suggested to determine the size and position of chromatin loops in meiosis (Schalbetter et al., 2019). The patterns of positioned loops in mitosis and meiosis are different, likely reflecting differences in the chromosomal structure and function in these two different yeast cell types.

Discussion section. Similarly, this finding was also demonstrated, clearly, by the meiotic yeast data, and thus should be described and cited.

We added: A similar omnipresence of positioned loops is also observed in yeast meiosis, and in mammalian cells (Schalbetter et al., 2019; Hsieh et al., 2020; Krietenstein et al., 2020; Rao et al., 2014).

Discussion section: As above, the conclusion that cohesin defines the loops is not entirely novel and previous work showing this should be cited here.

We added: The data from this and other studies in yeast revealed two differences between loop determinants in yeast and mammalian cells (Schalbetter et al., 2019; Dauban et al., 2020).

Discussion section. The model that is described is entirely congruent with the finding of the simulations employed by Schalbetter, 2019, which employed a stochastic model of loop expansion driven by extruders that could be blocked by barriers at preferred cohesin-binding sites (Rec8 in this case). As such, it would be entirely relevant to mention the congruence of the author's data (and model) with the findings of the polymer simulations developed by others.

We added: A similar model has been suggested for yeast meiotic chromosomes through polymer simulation modeling, and for mammalian mitotic chromosomes through an ingenious approach that allows differentiating contacts within and between sister chromatids (Schalbetter et al., 2019; Mitter et al., 2020).